# CaughtCheating: Is Your MLLM a Good Cheating Detective? Exploring the Boundary of Visual Perception and Reasoning

## Abstract

Recent agentic Multi-Modal Large Language Models (MLLMs) such as GPT-o3 have achieved near-ceiling scores on various existing benchmarks, motivating a demand for more challenging test tasks. These MLLMs have been reported to excel in a few expert-level tasks for humans, e.g., GeoGuesser, reflecting their potential as a detective who can notice minuscule cues in an image and weave them into coherent, situational explanations, leading to a reliable answer. But *can they match the performance of excellent human detectives?* To answer this question, we investigate some hard scenarios where GPT-o3 can still handle, and find a common scenario where o3's performance drops to nearly zero, which we name *CaughtCheating*. It is inspired by the social media requests that ask others to detect suspicious clues from photos shared by the poster's partner. We conduct extensive experiments and analysis to understand why existing MLLMs lack sufficient capability to solve this kind of task. CaughtCheating provides a class of challenging visual perception and reasoning tasks with great value and practical usage. Success in these tasks paves the way for MLLMs to acquire human-level detective perception and reasoning capabilities. The data and code are available at `https://anonymous.4open.science/r/CaughtCheating-0573/`.

## 1 Introduction

Recently advanced Multi-Modal Large Language Models (MLLMs) or corresponding Agents, such as GPT-o3 (OpenAI, 2025a) and Gemini-2.5 Pro (DeepMind, 2025a), have demonstrated extraordinary visual perception and reasoning capabilities (Yue et al., 2024b; Zhang et al., 2024a; Wang et al., 2024c; Chen et al., 2024a;b).

Recent studies have demonstrated that MLLMs are even capable of addressing far more demanding challenges, e.g., GeoGuesser, estimating an image's geographic location (Luo et al., 2025; Huang et al., 2025a). These kinds of tasks represent scenarios that even humans cannot accomplish easily, which require detective-level capabilities. These findings raise an important question: *Do recent MLLMs truly acquire detective-level perception and reasoning capabilities? If so, what is the boundary of their competence?*

Motivated by Human's Last Exam (Phan et al., 2025), which contains dozens of extremely challenging tasks, we aim to explore and evaluate the boundary of the detective-level ability (Gu et al., 2023; Yuan et al., 2025; de Lima et al., 2025) of MLLMs on visual perception and reasoning tasks. We investigate a number of hard scenarios where GPT-o3 can solve the queries even though they are challenging for humans. Then we discover a common scenario where o3's performance drops dramatically to almost the random guess level. This scenario is inspired by the social media requests that ask others to detect potential suspicious clues from photos shared by the poster's partner, which go against the partner's claims. Figure 1 shows an example, in which the user query is: "*My boyfriend said he's dining alone at the restaurant and sent me this photo. Do you notice anything suspicious in this image that contradicts his claim?*" This image itself seems an ordinary food-sharing image, while in the reflection of the spoon, there are other people, including a girl with long hair, who can be visible, which is suspicious and violates the claim of being alone. For this

kind of task, we find that most humans, and the strong MLLMs like o3, are not able to identify the clues, indicating the superior detective-level capabilities required.

Thus, to explore the boundary of the visual perception and reasoning capabilities of current MLLMs (Johnson et al., 2017; Zellers et al., 2019; Chen et al., 2024a;b), we collect these images and construct the CaughtCheating benchmark. This benchmark consists of a *Real Subset* and a *Synthetic Subset*. The real subset consists of 100 mannully screened images[1] sourced from publicly posted photographs on social media. The dataset is nearly evenly split into a *Clued* category and a *Unclued* category. Annotations for each image include a primary question about potential violation of the original claims, corresponding deterministic and non-deterministic clues, and a series of decomposed questions to analyze the visual reasoning process of MLLMs. The synthetic subset consists of 3700 verified synthetic images generated by GPT-Image-1 with diverse scenes. The two subsets are designed to be *complementary to each other*, as the real subset represents the extremely rare and challenging real-world cases, while the synthetic subset represents less challenging but more diverse cases. Most of the analysis is conducted on the real subset.

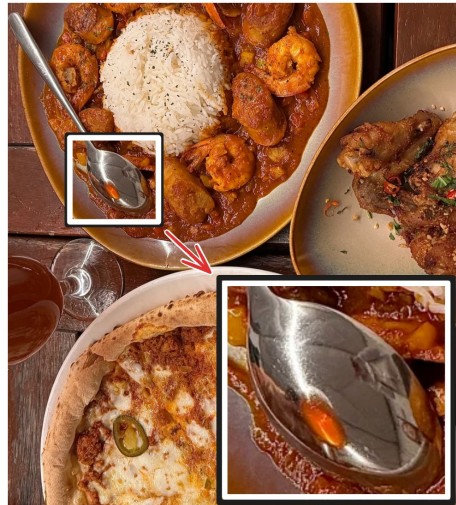

Figure 1: An example from CaughtCheating. **Query**: "*My boyfriend said he's dining alone at the restaurant and sent me this photo. Do you notice anything suspicious in this image that contradicts his claim?*" **Suspicious Clue**: "*There are other people, including a girl with long hair, visible in the spoon's reflection.*" In this example, most human participants, and the strong o3 are not capable of identifying this clue when not given any hints, indicating the demandforf superior detective-level capabilities.

CaughtCheating is more challenging than all the previous tasks because the targets to be identified are not directly defined in the query, and thus can not be solved by an exhaustive grid search. For example, when o3 tries to solve the query in Figure 1, it conducts the exhaustive grid search by focusing on one part of the figure at a time. However, even if it has tried focusing its attention on the area with the spoons, it still can not find this clue[2]. To theoretically analyze the difficulty dependencies between CaughtCheating and existing challenging tasks and understand the reasons behind the failures of o3, we introduce the *Guided Search* theory from cognitive science (Wolfe et al., 1989; Itti & Koch, 2001; Itti et al., 2002; Duncan & Humphreys, 1989) and the factors that guide attention in visual search. According to the theory, CaughtCheating has low bottom-up salience, lacks top-down feature guidance, and contains blurry scene structure and meaning.

Extensive evaluation results demonstrate that current MLLMs perform poorly on our detection-level benchmark of CaughtCheating. Notably, on the *Real Subset*, even the best-performing model (o3) achieved only 26.0% accuracy in detecting the deterministic clues hidden in the images and 17.2% IoU (the intersection over union). Moreover, the accuracy of justifying the absence of suspicious clues (*Unclued Acc*) is only 8.0%, resulting in the overall F1 score being only 23.9%. Through investigation, we find that the current advanced MLLMs, e.g., o3 and Gemini-2.5-pro, **not only fail to identify the deterministic clues, but also tend to hallucinate and accuse everything by generating lots of so-called suspicious clues, even for innocent images**, which is not preferred. Taken together, these results show the significance of CaughtCheating, which reveals that recent MLLMs still lack detective-level capabilities, and further exposes the current boundary of their visual perception and reasoning capabilities. Our contributions can be summarized as:

- We systematically evaluate the limits of current MLLMs in visual perception and reasoning, analyzing how they solve various complex tasks via sophisticated reasoning strategies, and identify a common scenario where even advanced models like o3's performance drops to nearly zero.

---

[1]This kind of data is intrinsically scarce. We have manually screened and verified almost all the existing related posts on public social media to construct this benchmark.

[2]o3's visual reasoning traces are presented in Appendix E.

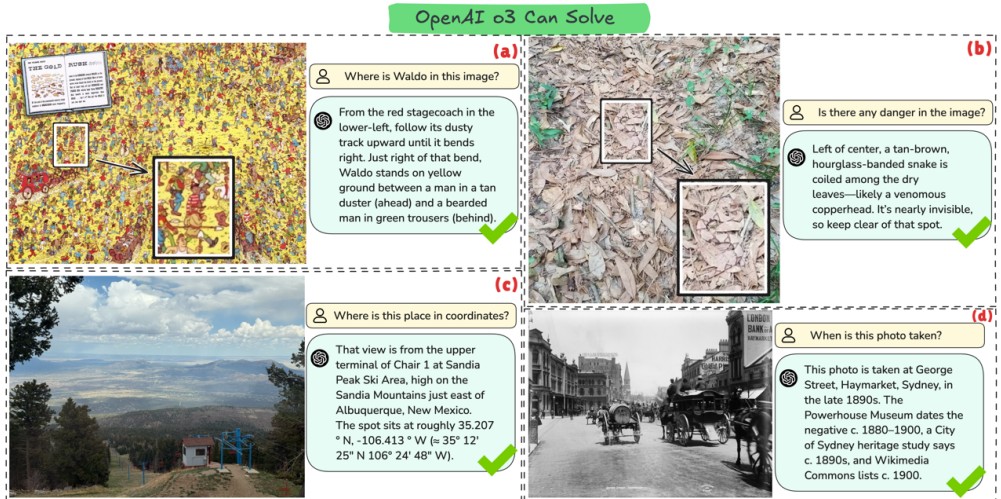

Figure 2: **Demonstration of GPT-o3's multimodal visual-reasoning breadth.** (a) *Visual search*: locating Waldo in a densely populated illustration. (b) *Visual search for camouflage*: spotting a nearly invisible copperhead snake hidden among dry leaves. (c) *GeoGuessr*: identifying the upper terminal of Chair 1 at New Mexico, and estimating its latitude/longitude from a single image. (d) *TimeGuesser*: dating the photograph by matching architectural signage and period vehicles to museum and heritage records. These examples highlight o3's strong visual perception and reasoning capacity across various visual tasks that most humans can not accomplish.

- We present CaughtCheating, the first benchmark specifically designed to assess the ability to actively search and detect subtle, context-dependent suspicious clues in real-world images. Most human annotators and state-of-the-art agentic MLLMs struggle to succeed on CaughtCheating tasks, highlighting the lack of detective-level exploration skills.

- We analyze why even the most advanced agentic MLLMs fail on CaughtCheating. Inspired by the *Guided Search* theory, we find that these models often lack awareness of *what to search for* and *how to relate observed details to the query*. Our findings offer insights into both the construction of more challenging benchmarks and the limitations of existing MLLMs.

## 2 EXPLORING THE BOUNDARY OF VISUAL PERCEPTION AND REASONING

### 2.1 REASONING TRACE ANALYSIS OF O3

As shown in the Figure 2, 4 representative task scenarios are selected for our qualitative analysis towards the boundary of MLLM visual perception and reasoning capabilities. These tasks have been shown to be solved by the powerful agentic MLLM, GPT-o3, even if most of them can not be solved by individual humans[3].

When solving (a), o3 systematically sweeps the image from broad overviews to focused zooms, homing in on red-and-white horizontal stripes of the character "Waldo". After eliminating false matches quadrant by quadrant, it confirms Waldo's outfit and hat, then translates his pixel coordinates into an easy landmark description. When solving (b), the o3 methodically zooms into different areas of the leaf-litter image, from the center, lower left, and lower right, to searching for irregular shapes or patterns. Spotting rounded tan-brown coils with dark hourglass bands just left of center, then it recognizes the tell-tale camouflage of a venomous pit viper (likely a copperhead). When solving (c), o3 compares visual clues in the photo, red chairs on blue lift towers, the wide west, facing vista over Albuquerque's grid, and the tree-line/elevation typical of Sandia Crest, with known features of Sandia Peak Ski Area. Cross-checking those details against published coordinates confirms the match. When solving (d), o3 cross-checks catalog records for Henry King's glass-plate negatives with heritage reports that caption this very view "c. 1890s." Then it matches visual clues, horse buses and a Sydney Municipal, dense telegraph wires but no electric-tram overhead, and the original Anthony Hordern's "Palace Emporium" sign that vanished after the 1901 fire, to pin the scene to the year.

---

[3]All the screenshots of o3 reasoning traces for solving these examples are provided in the Appendix E.

According to the above analysis, we find that the o3 model approaches these tasks with a **methodical, exhaustive grid search**, inspecting each region or object one by one until all plausible candidates are ruled in or out. However, the effectiveness of this exhaustive approach will be largely negatively affected if the target object is easily overlooked. Figure 1 presents an example: When trying to solve the given query, o3 zooms in on the areas including pizza to confirm if slices were missing, the spoon and glass reflections to spot another diner, and the wing plate and surrounding dishes to gauge portion sizes and leftover clues. However, *it fails to notice that there are multiple people visible in the spoon's reflection.* Compared with other objects, the spoon is so negligible that o3 does not pay much attention to it, thus leading to the failure. Moreover, even occasionally, o3 coincidentally pays more attention to the spoon, it can not successfully perceive the content in the reflection. To conclude, we find that even though o3 is able to accomplish some complex tasks, it mainly relies on an exhaustive grid search, which indicates a lack of detective-level visual perception and reasoning capabilities.

## 2.2 GUIDED SEARCH THEORY

To theoretically analyze the differences between the existing visual tasks and CaughtCheating, we introduce the *Guided Search* theory (Wolfe et al., 1989) and the corresponding factors (Wolfe & Horowitz, 2017) that guide attention in visual search in the area of cognitive science. In its theory, searching involves directing attention to objects that might be the target. This process is guided to the most promising items and locations by five factors discussed in the theory: **bottom-up salience**, **top-down feature guidance**, **scene structure and meaning**, **the previous history of search**, and **the relative value of the targets and distractors**. Through investigation on the reasoning traces of o3, we find this theory, though initially proposed in the area of cognitive science, is still applicable to the current MLLMs. We argue that CaughtCheating is significantly more challenging than many existing visual reasoning tasks, including those depicted in Figure 2, due to the interplay of these factors.

**Bottom-Up Salience** refers to the extent to which an item "pops out" from its surroundings due to its intrinsic visual properties (e.g., color, orientation, contrast). This aspect represents the easiest strategy to make visual search hard. In examples like Figure 2 (a) and (b), both the targeting objects have low bottom-up salience, making them hard to find and requiring exhaustive searches. Similarly, suspicious cues in CaughtCheating also have *extremely low bottom-up salience*, like a reflection in a spoon, a partially obscured object, or a subtle item in the background, and are easily overlooked.

**Top-Down Feature Guidance** involves using knowledge about the target's properties to guide search. Previous tasks benefit significantly from top-down guidance. For Waldo, the model searches for specific red-and-white stripes as a distinct character. For the snake, the query about "danger" might guide the model to look for threatening patterns. GeoGuesser and TimeGuesser rely on identifying specific architectural styles, vegetation, or period-specific artifacts. However, this is where CaughtCheating poses a major hurdle. The "target", i.e., the suspicious clue, is often *not a predefined object but an anomaly whose significance is context-dependent.* Lacking the top-down feature guidance, the model *does not know what to look for* because the clue could be almost anything (an extra glass, a reflection, an out-of-place item). As observed, even if o3 occasionally focuses on the correct object (like the spoon), it may still fail to perceive the detail within it or infer its implication.

For **Scene Structure and Meaning**, the understanding of typical scene layouts and the relationships between objects helps guide attention to likely target locations. For previous tasks, o3 leverages scene context effectively. In GeoGuesser, it compares visual clues with known features of geographical locations. In TimeGuesser, it matches visual clues like vehicles and signage to historical records. However, for CaughtCheating, the image itself might seem like an ordinary food picture or a hotel picture. Allocating the critical visual clues for the task does not merely require object recognition; it also needs to interpret subtle social cues and deviations from a presumed norm (e.g., "dining alone"). Current MLLMs struggle with this divergent reasoning over subtle, context-dependent cues, often focusing on non-deterministic details rather than decisive evidence.

In summary, CaughtCheating is more complex due to the extremely low bottom-up salience of crucial cues, the profound lack of specific top-down feature guidance, and the need to interpret subtle social context rather than just recognizing objects or well-defined patterns. While current agentic MLLMs can methodically search and identify objects through a process of elimination and feature matching, CaughtCheating demands a more nuanced "detective-level" ability to identify initially inconspicuous details and infer their significance within a specific social claim.

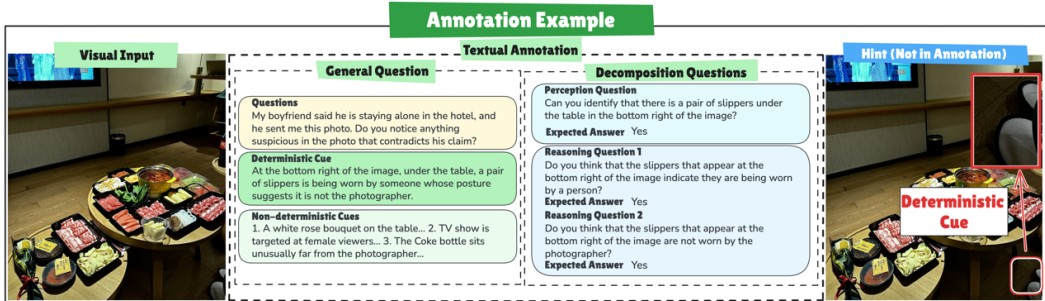

Figure 3: **An example of the annotation for the "Clued" category.** Each image is annotated with a general question assessing overall suspicion and decomposed questions focused on a deterministic clue (here, the feminine bow hair accessory). Decomposed questions include perception-based inquiries (clue identification) and reasoning-based inquiries (social implications and contradictions), all annotated with the expected answer "yes".

## 3 BENCHMARK CONSTRUCTION

### 3.1 IMAGE COLLECTION FOR REAL SUBSET

We collect images from publicly posted photographs on social media, focusing on those posts that request others to detect potential suspicious clues that violate their partners' claims from the photos. We only collect images that either clearly contain or lack subtle clues related to potential violation of the claim. Each image is manually reviewed to ensure sufficient resolution quality for identifying such clues. Due to the limited availability of images with naturally occurring subtle clues, we apply minimal cropping to some images that originally show multiple people, transforming them into single-person photos while preserving subtle indicators of another person's presence. This approach allows us to create challenging cases where the clues are interpretable for humans but not immediately obvious. To ensure practical relevance, we exclude any synthetic images generated by image generation models. After careful selection and verification, we construct a dataset of 100 images, split into *Clued* and *Unclued* categories, with all personal information removed. A detailed version of *Benchmark Construction*, including the image examples, is provided in the Appendix B.

### 3.2 ANNOTATION FOR REAL SUBSET

After constructing the image set, we annotate each image with a set of questions and corresponding ground-truth answers. A detailed annotated example is shown in Figure 3. For images in *Clued* category, we annotate each one using a question instantiated from the template: "*My [girlfriend/boyfriend] said [she/he] is [in a certain scenario] and sent me this photo. Do you notice anything suspicious in the image that contradicts [her/his] claim?*" Among the potential clues, the one that deterministically shows the violation of the providing claim (a clearly identifiable, contextually inappropriate element) will be selected as the ***Deterministic Clue***, e.g., a pair of slippers is being worn by someone in Figure 3. The remaining clues are labeled as ***Non-deterministic Clues*** (weaker or more ambiguous signals), e.g., the rose bouquet, the TV shows, and the far-reached drinks. These non-deterministic clues might be suspicious, but apparently not enough to infer the potential claim violation. The reason we provide these clues is to avoid punishing models when they mention these clues.

Furthermore, we construct a series of decomposed questions designed to analyze the visual reasoning process of MLLMs, shown in the right part of Figure 3. This series includes: (1) ***Decomposed Perception Question***, which assesses whether the MLLMs can identify the deterministic clue when we explicitly mention the clue and position. (2) ***Decomposed Reasoning Question***, which assesses whether MLLMs can understand the social implications of the clue, or whether MLLMs can imply the relation between the clue and the potential cheating. The correct answer to each of these decomposed questions is annotated as "yes". We annotate each image in the *Unclued* category using the same initial question template, with "There is no clear evidence." as the ground-truth answer.

### 3.3 Construction for Synthetic Subset

As a complement to the real subset, we further construct the synthetic subset through a cascading pipeline that balances quality and diversity. First, we define a set of scenes (e.g., bar, office, hotel room, car) and introduce deterministic clues in complete sentences (e.g., There are two glasses on the table) anchored in scene context and gender stereotypes, while ensuring gender balance. Then, for clued samples, the LLM is given both the scene and its associated clue to generate first-person photo prompts that contain elements in contradiction to the claim. For unclued samples, only the scene is provided, and the LLM is instructed to generate first-person photo prompts of a single individual without such contradiction. Candidate prompts are then reviewed by annotators for semantic accuracy and compliance, with the best five prompts per clue retained.

Subsequently, we employ a state-of-the-art image generation model (GPT-Image-1 (OpenAI, 2025b)) to synthesize multiple candidates per prompt, from which human evaluators select the best five. Question templates are adapted from the real subset, and answers are directly derived from clue sentences, ensuring consistency with annotation protocols. This procedure yields 3700 images, evenly divided into clued and unclued categories across 10 scenes. Detailed method, including mappings between scenes and clues, prompt design, generation workflow, and distributional statistics, is provided in Appendix B.

### 3.4 Evaluation Metrics

We employ several evaluation metrics to comprehensively assess MLLMs' performance in detecting potential claim violations from images. **Clued Accuracy (Clued Acc)** measures whether MLLMs can successfully identify the key deterministic clues in images from the *Clued* category. **Intersection over Union (Clued IoU)** evaluates how well MLLMs identify all relevant non-deterministic clues while avoiding unrelated elements in the *Clued* category. **Unclued Accuracy (Unclued Acc)** assesses whether MLLMs can correctly determine the absence of suspicious clues in images from the *Unclued* category. In addition to the above three metrics, we also report the accuracy of MLLMs on the decomposed questions in the analysis, including *Decomposed Perception Accuracy (Dec. P Acc)*, *Decomposed Reasoning Accuracy (Dec. R Acc)*, and *Decomposed Overall Accuracy (Dec. Acc)* for in-depth analysis. These metrics together provide a comprehensive evaluation framework that captures both the accuracy of clue detection and the quality of reasoning in different scenarios.

To compute these metrics, we need to parse the key points from MLLMs' open-ended responses and compare them with the ground-truth answers. Given the complexity of this task and the diversity of the responses, we recommend using human evaluators as the primary judges for the most accurate assessment. However, to enable fair and automated comparison across different models, we also develop an automatic evaluation approach using GPT-4.1 to parse and compare the model response. To validate the reliability of our automatic evaluation method, we calculate the inter-rater agreement between human evaluators and GPT-4.1 using Cohen's Kappa coefficient. The resulting kappa scores of 0.82 for *Clued Acc* and 0.943 for *Unclued Acc* demonstrate strong alignment between human and automatic evaluation, indicating the reliability of our automated assessment approach. A detailed version of *Evaluation Metrics*, including the calculation and transformation between metrics, is provided in the Appendix C.

## 4 Experimental Results

### 4.1 Main Results

The main results are shown in Table 1. *Clued Acc* and *Clued IoU* represent the capability of MLLMs to identify the suspicious clues, which directly reflect the MLLMs' visual perception and reasoning abilities. For previous open-source models, the performance is almost negligible, as no models can reach an accuracy above 5%, indicating their inferior capabilities on visual perception, reasoning, or even instruction following. As for proprietary models before the reasoning era, GPT-4o achieves 4.0% accuracy and 1.0% IoU, and Gemini-2-flash achieves 10.0% accuracy and 0.0% IoU. The performances are slightly better, indicating their better capabilities in instruction understanding and following, but still they can not reach accuracies above 10%. Only for the recent strong large reasoning models, like Gemini-2.5-pro and GPT-o3, the performances can reach above 20% accuracy

| | Synthetic Subset | | | Real Subset | | | | | |
| | Overall | | | Clued | | Unclued | Overall | | |
| | Precision ↑ | Recall ↑ | F1 ↑ | Acc ↑ | IoU ↑ | Acc ↑ | Precision ↑ | Recall ↑ | F1 ↑ |
|---|---|---|---|---|---|---|---|---|---|
| **InternVL2-1B** (Chen et al., 2024e) | 15.6 | 2.3 | 4.0 | 0.0 | 0.0 | 82.0 | 0.0 | 0.0 | 0.0 |
| **LLaVA-OV-1B** (Li et al., 2024) | 33.7 | 3.2 | 5.8 | 0.0 | 0.0 | 86.0 | 0.0 | 0.0 | 0.0 |
| **InternVL2.5-1B** (Chen et al., 2024e) | 23.2 | 3.5 | 6.1 | 0.0 | 0.0 | 94.0 | 0.0 | 0.0 | 0.0 |
| **InternVL2-2B** (Chen et al., 2024e) | 27.1 | 5.6 | 9.3 | 0.0 | 0.0 | 76.0 | 0.0 | 0.0 | 0.0 |
| **InternVL2.5-2B** (Chen et al., 2024e) | 26.3 | 4.5 | 7.7 | 0.0 | 0.0 | 68.0 | 0.0 | 0.0 | 0.0 |
| **Qwen2.5-VL-3B** (Bai et al., 2025) | 52.2 | 37.6 | 43.7 | 2.0 | 0.0 | 50.0 | 3.8 | 2.0 | 2.6 |
| **LLaVA-v1.6-Mistral-7B** (Li et al., 2024) | 69.7 | 7.6 | 13.7 | 0.0 | 0.0 | 82.0 | 0.0 | 0.0 | 0.0 |
| **LLaVA-OV-7B** (Li et al., 2024) | 88.7 | 10.2 | 18.3 | 2.0 | 0.0 | 52.0 | 4.0 | 2.0 | 2.7 |
| **Qwen2.5-VL-7B** (Bai et al., 2025) | 62.6 | 45.6 | 52.8 | 2.0 | 3.9 | 66.0 | 5.6 | 2.0 | 2.9 |
| **InternVL2-8B** (Chen et al., 2024e) | 44.2 | 7.6 | 13.0 | 0.0 | 0.0 | 76.0 | 0.0 | 0.0 | 0.0 |
| **InternVL2.5-8B** (Chen et al., 2024e) | 65.4 | 34.2 | 44.9 | 0.0 | 0.0 | 72.0 | 0.0 | 0.0 | 0.0 |
| **LLaVA-1.6-Vicuna-13B** (Li et al., 2024) | 20.5 | 3.3 | 5.7 | 0.0 | 0.0 | 72.0 | 0.0 | 0.0 | 0.0 |
| **InternVL2-26B** (Chen et al., 2024e) | 61.0 | 33.7 | 43.4 | 2.0 | 1.8 | 10.0 | 2.2 | 2.0 | 2.1 |
| **InternVL2.5-26B** (Chen et al., 2024e) | 73.8 | 38.9 | 50.9 | 0.0 | 0.0 | 80.0 | 0.0 | 0.0 | 0.0 |
| **InternVL2.5-38B** (Chen et al., 2024e) | 65.5 | 39.5 | 49.3 | 2.0 | 0.0 | 76.0 | 7.7 | 2.0 | 3.2 |
| **InternVL2-40B** (Chen et al., 2024e) | 71.8 | 37.6 | 49.3 | 4.0 | 0.7 | 12.0 | 4.4 | 4.0 | 4.2 |
| **InternVL2-72B** (Chen et al., 2024e) | 88.4 | 46.3 | 60.8 | 4.0 | 0.8 | 16.0 | 4.5 | 4.0 | 4.3 |
| **InternVL2.5-72B** (Chen et al., 2024e) | 75.5 | 44.7 | 56.2 | 2.0 | 0.8 | 80.0 | 9.1 | 2.0 | 3.3 |
| **LLaVA-OV-72B** (Li et al., 2024) | 77.6 | 42.3 | 54.8 | 0.0 | 1.3 | 72.0 | 0.0 | 0.0 | 0.0 |
| **GPT-4o** (OpenAI et al., 2024) | **81.1** | 53.2 | 64.2 | 4.0 | 1.0 | 54.0 | 8.0 | 4.0 | 5.3 |
| **Gemini-2-flash** (DeepMind, 2025b) | 71.4 | 50.0 | 58.8 | 10.0 | 0.0 | 6.0 | 9.6 | 10.0 | 9.8 |
| **Gemini-2.5-flash** (DeepMind, 2025a) | 71.1 | 84.5 | 77.2 | 18.0 | 5.1 | 22.0 | 18.8 | 18.0 | 18.4 |
| **Gemini-2.5-pro** (DeepMind, 2025a) | 72.6 | 87.2 | 79.2 | 20.0 | 15.1 | 22.0 | 20.4 | 20.0 | 20.2 |
| **GPT-o3** (OpenAI, 2025a) | 68.6 | **94.7** | **79.6** | **26.0** | **17.2** | 8.0 | **22.0** | **26.0** | **23.9** |
| **Human** | 97.0 | 95.8 | 96.4 | 56.0 | / | 68.0 | 63.6 | 56.0 | 59.6 |

Table 1: The overall precision, recall, and F1 score for the *Synthetic Subset*, and the accuracies, IoU on the *Clued* category, and the accuracy on the *Unclued* category for the *Real Subset*, and the overall precision, recall, and F1 score for the *Real Subset*. Models are grouped by parameter size and type (open-source vs. proprietary). *Clued Acc* and *IoU* represent the capability of MLLMs to identify the suspicious clues, which directly reflects the MLLMs' visual perception and reasoning abilities. *Unclued Acc* represents the capability of MLLMs to not generate any suspicious clues if the image is unclued. On the *Real Subset*, even the best performing model, *GPT-o3*, only achieves 26.0% accuracy and 17.2% IoU, indicating the current boundary of MLLMs' capabilities. *F1* score shows the overall capability of MLLMs on CaughtCheating, where *GPT-o3*, achieves only 23.9%. The highest *F1* score is 23.9%, which is much lower than the human performance, indicating the current boundary of MLLMs' capabilities.

and 10% IoU, indicating their strong capabilities on visual perception and reasoning. But still, even the best performing model, *GPT-o3*, only achieves 26.0% accuracy and 17.2% IoU, **indicating the current boundary of MLLMs' capabilities.** We believe this benchmark is challenging enough and shows the current boundary of their visual perception and reasoning capabilities.

In the meantime, we also report the *Unclued Acc* to evaluate the capability of MLLMs to not generate any suspicious clues if the image is unclued. This is also important for the real-world application, as **we do not prefer MLLMs to suspect and accuse anything if the image providers are innocent**. As shown in the table, most of the models reach high accuracies on this category; however, this performance is due to their inability to generate any suspicious clues. On the contrary, the advanced agentic models, Gemini-2.5-pro and GPT-o3, achieve low accuracies on this category, indicating their hallucination of nonexistent suspicious clues even on unclued images. These low accuracies reveal **their lack of strong reasoning abilities to identify if something is suspicious or not.**

Finally, the *F1* scores represent the overall performance of the model, which is the harmonic mean of the *Precision* and *Recall*. The highest *F1* score is 23.9%, which is much lower than the human performance, indicating the current boundary of MLLMs' capabilities.

Alternatively, on the *Synthetic Subset* with 3,700 images, though previous models still perform poorly, existing state-of-the-art MLLMs like GPT-o3 and Gemini-2.5-pro can reach F1 scores of almost to 80%. The main reason is that the current image synthesis model, GPT-Image-1, is not able to generate scenes with subtle and non-obvious suspicious clues, e.g., the vague reflections in the mirror. On the contrary, current SOTA MLLMs can already have the ability to identify the clues generated by image synthesis models. Further discussion on the quality of synthetic data is provided in Appendix B.

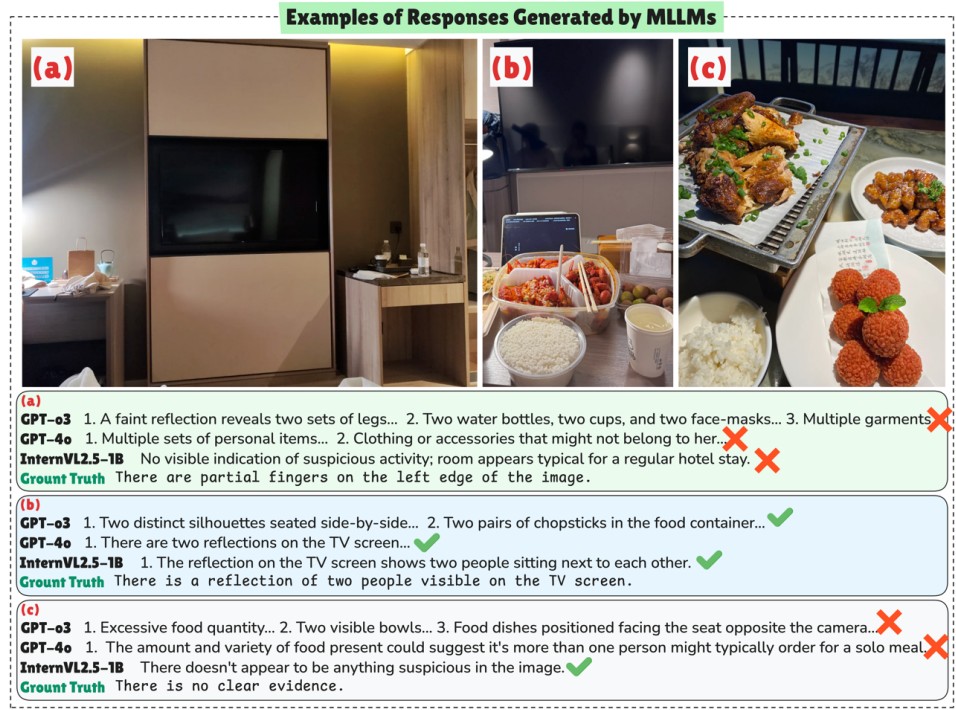

Figure 4: **Case studies of the models' performance on the CaughtCheating examples.** 3 representative models are selected, including GPT-o3, GPT-4o and InternVL2.5-1B, and 3 images are selected: (a) A difficult *Clued* image, (b) An easy *Clued* image, and (c) An *Unclued* image. The models' responses are truncated for better visualization. High-resolution versions of each subfigure are provided in the Appendix F.

## 4.2 DECOMPOSITION ANALYSIS

To better understand why the current advanced MLLMs can not perform well in this task, we design a set of decomposed questions for each image in the *Clued* category. These questions are divided into two types: perception questions, which test whether the model can accurately identify the key deterministic clue when it is explicitly mentioned, and reasoning questions, which assess whether the model can correctly infer the implications or contradictions associated. This fine-grained analysis helps reveal whether a model's failure is due to not seeing the clue at all, or seeing it but not understanding its significance, thus providing deeper insight into the limitations.

|  | Dec. P | Dec. R | Dec. | Clued ↑ |
|---|---|---|---|---|
| **GPT-4o** | 52.0 | 12.8 | 2.0 | 4.0 |
| **Gemini-2-flash** | 74.0 | 69.6 | 38.0 | 10.0 |
| **Gemini-2.5-flash** | 72.0 | 39.2 | 20.0 | 18.0 |
| **Gemini-2.5-pro** | 80.0 | 52.9 | 34.0 | 20.0 |
| **GPT-o3** | 62.0 | 24.5 | 2.0 | **26.0** |
| **Human** | 82.0 | 97.8 | 80.0 | 56.0 |

Table 2: **Performance on decomposed questions.** *Dec. P* and *Dec. R* is the *Decomposed Perception Accuracy* and *Decomposed Reasoning Accuracy*. *Dec.* is the *Decomposed Accuracy*, i.e., proportion of the model correctly answering all decomposed questions.

As shown in Table 2, the *Dec. P* is far higher than the *Clued Acc*, indicating that the models can identify the key deterministic clue when it is explicitly mentioned. When the human participants are given the image, it's hard for them to identify the suspicious clues at the first place, e.g. the refelction in Figure 1 and the femine bow hair in Figure 3, but once they are explicitly mentioned or pointed out, they will admit the presense of the items. This human behavior leads to the relatively high *Dec. P* for humans. For the *Dec. R*, the performances are all relatively lower, especially for GPT-4o and o3.

These results together indicate that current advanced MLLMs can identify the key subtle items in the image if they are explicitly mentioned. However, in CaughtCheating, when being asked to identify the suspicious clues without being given any hints, they tend to do an exhaustive search and generate

lots of clues without really judging if the clues are suspicious or not, and at the same time, ignore the key but subtle deterministic clues. These behaviors are similar to humans and verify the hypothesis based on *Guided Search* theory.

## 5 CASE STUDIES

In this section, we provide some examples to show how exactly different models perform on CaughtCheating, shown in Figure 4. In the figure, 3 representative models are selected, including GPT-o3, GPT-4o, and InternVL2.5-1B, and 3 images are selected: (a) A difficult *Clued* image, (b) An easy *Clued* image, and (c) An *Unclued* image.

In (a), there is an elbow, and fingers are visible at the left edge of the photo, clearly indicating the presence of another person. However, all the models fail to identify this subtle but deterministic clue and focus on the reflection of the television, even though there are no visible clues in the reflection, as another person is sitting by the table. What's worse, o3 and 4o keep mentioning the two bottles or cups, which are obviously provided by the hotel and can not be the suspicious clues. On the contrary, InternVL2.5-1B can not provide any clues by saying this is a normal hotel image. In (b), the reflection in the TV clearly shows there are two people on the bed, thus all the selected models can identify this clue. These 2 examples, (a) and (b), show that: (1) models are able to see through reflections, and (2) Reflection does not always contain suspicious clues, which further verifies that **CaughtCheating is challenging since there are no fixed rules for the suspicious clues.**

(c) shows an *Uncled* image, which is merely an ordinary food-sharing image. However, o3 still tries to generate a lot of *so-called suspicious clues*, including the amount of food, the place settings, etc. **This behaviour is not expected since we only want models to generate clues that are really suspicious, rather than accusing everything, which further indicates the values of CaughtCheating.** Similarly to the above examples, InternVL2.5-1B can not provide any clues by saying this is a normal food-sharing image, that's why it reaches the highest on the *Unclued Acc*.

## 6 CONCLUSION

In this work, we introduce CaughtCheating, a benchmark that stress-tests detective-level visual perception and social reasoning by asking models to find subtle, context-dependent clues that contradict a subject's claim. Grounded in Guided Search theory, our analysis reveals why these cases are uniquely challenging—targets exhibit extremely low bottom-up salience, lack top-down feature guidance, and hinge on scene meaning, making exhaustive scan strategies ineffective. The benchmark combines a carefully curated real subset and a complementary synthetic subset with a validated automatic evaluator, enabling reproducible, fine-grained assessments. Results on the real subset reveal stark limitations: even strong agentic MLLMs significantly underperform—GPT-o3 attains only 26.0% clued accuracy, 17.2% IoU, 8.0% unclued accuracy, and an overall F1 of 23.9%. As society continuously expects more sophisticated capabilities from LLMs, we anticipate that real-world complex tasks requiring detective-level abilities will become increasingly important for MLLMs to master. In summary, CaughtCheating not only provides a challenging new benchmark for evaluating multimodal reasoning, but also exposes critical weaknesses in current MLLMs, highlighting the urgent need for further research into the detective-level capabilities of MLLMs.

## 7 ETHICS CONCERNS AND LIMITATIONS

Because our benchmark relies exclusively on publicly available for the *Real Subset*, annotatable social-media photographs, the source pool overwhelmingly features cisgender, heterosexual couples. The scarcity of labeled images depicting LGBTQ+ or non-monogamous relationships, therefore compelled us to centre this demographic. The same constraint restricted the dataset to a handful of commonplace settings, such as hotels, restaurants, cafés, and vacation scenes, leaving contexts such as nightlife, workplaces, or culturally specific environments underrepresented. The benchmark addresses a single, socially charged form of visual reasoning, detecting suspected infidelity, rather than the broader universe of complex social-reasoning inferences people may draw from images. These limitations stem from the paucity of suitable public data. To alleviate these concerns, we further construct the synthetic subset through a cascading pipeline that balances quality and diversity. Although the synthetic subset is not as challenging as the real-world scenarios, it alleviates the potential ethical concerns and expands the dataset to a more diverse set of scenes.

## REPRODUCIBILITY STATEMENT

All the prompts for evaluation, prompts for generating the synthetic data, along with the scene distribution, have been provided in the Appendix B. Moreover, all the code and data are available at `https://anonymous.4open.science/r/CaughtCheating-0573/`.

## THE USE OF LARGE LANGUAGE MODELS

Large Language Models were utilized solely as writing aids during the preparation of this manuscript. These models assisted with enhancing sentence clarity, improving readability, and maintaining stylistic coherence throughout the text. All suggestions provided by the LLMs were thoroughly examined, edited, and refined by the authors before incorporation. The LLMs played no role in conceptual development, experimental design, data interpretation, or the formulation of scientific insights. The entirety of the research contributions, analytical interpretations, and conclusions presented herein is the original work of the authors.

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

TABLE OF CONTENTS FOR APPENDIX

## A RELATE WORK

### A.1 LLM REASONING

The chain-of-thought technique (Wei et al., 2022; Kojima et al., 2022) represents the early efforts in exploring the reasoning capabilities of large language models (LLMs) (Brown et al., 2020; Chowdhery et al., 2023). By explicitly generating intermediate reasoning steps, this method notably enhances performance across various reasoning tasks (Patel et al., 2021; Cobbe et al., 2021). Moreover, advances in decoding strategies have introduced additional test-time computation to further boost performance. For instance, Self-Consistency sampling (Wang et al., 2022b), which employs voting mechanisms to select from multiple reasoning paths, has notably increased reliability. Expanding beyond linear reasoning processes, structured frameworks such as Tree-of-thought Yao et al. (2023) or Graph-of-thought (Jin et al., 2024)facilitate the exploration of multiple candidate reasoning paths within branched subspaces before reaching a final conclusion. Other research investigates manipulating the reasoning process to generate longer chains of thought than those typically observed, either by explicitly prompting extended reasoning chains (Muennighoff et al., 2025) or by integrating human-like cognitive theory foundations into the inference process (Zhou et al., 2023; Gandhi et al., 2023; Lee et al., 2024; Chen et al., 2025). Furthermore, supervised fine-tuning (SFT) not only improves general instruction-following performance (Ouyang et al., 2022; Xia et al., 2024) but has also been demonstrated to significantly enhance multi-step reasoning capabilities when trained on structured chain-of-thought (CoT) traces, where models learn to explicitly generate intermediate reasoning steps (Ranaldi & Freitas, 2024; Wen et al., 2025; Muennighoff et al., 2025; Yu et al., 2025; Li et al., 2025a). Additionally, prior research has employed reward models during training to evaluate each intermediate reasoning step individually, rather than solely assessing final outcomes, further improving reasoning performance (Uesato et al., 2022; Lightman et al., 2023). This approach integrates effectively with Monte Carlo Tree Search techniques (Xie et al., 2024), providing valuable insights into performance gains achieved through fine-grained value estimations. Beyond training, many studies augment the reasoning process with the ability to invoke external tools and knowledge sources, a paradigm known as "agentic reasoning" (Wu et al., 2025). In this paradigm, LLMs call tools such as calculators, code interpreters, web search, and other utilities to provide context from the tools' results into the reasoning process to solve complex tasks. For instance, Jin et al. (Jin et al., 2025) introduce the Search-R1, which lets an LLM query a search engine and condition subsequent reasoning on the retrieved evidence. Recent developments in large-scale reinforcement learning, relying solely on outcome-based rewards, have demonstrated potential for inducing emergent multi-step reasoning capabilities (Guo et al., 2025; Jaech et al., 2024). While the advancements in reasoning also potentially lead to the emergence of overthinking issues (Chen et al., 2024c; Fan et al., 2025). Such advancements underscore the importance of tasks that can be automatically verified (e.g., RL can be effectively scaled up with minimal noise in its reward signals).

### A.2 MLLM REASONING

Recent developments in MLLMs (Wang et al., 2022a; Liu et al., 2023; OpenAI et al., 2024; Liu et al., 2024; Chen et al., 2024e;d; Bai et al., 2025) have led to the exploration of multimodal chain-of-thought techniques aimed at enhancing performance on visual reasoning tasks (Yu et al., 2023; Lu et al., 2023; Hao et al., 2025) with both textual reasoning process (Lu et al., 2022; Zhang et al., 2023) and multimodal reasoning path (Wu et al., 2024; Fu et al., 2025). Methods such as rationale distillation and self-reflection have also been employed to strengthen reasoning capabilities (Zhang et al., 2024b; Zhou et al., 2024; Wang et al., 2024a;b; Deng et al., 2024). Besides, LLaVA-o1 (Xu et al., 2024) proposes a fine-tuning strategy that leverages a dataset enriched with structured reasoning annotations (e.g., summarization, visual analysis, logical deduction, conclusion), achieving substantial performance improvement. Inspired by successes in reinforcement learning of LLMs, recent efforts have similarly applied this method to visual math problems and other visual question-answering tasks (Deng et al., 2025b; Huang et al., 2025b; Wang et al., 2025b; Peng et al., 2025; Meng et al., 2025). For example, Curr-ReFT (Deng et al., 2025a) introduces a three-stage progression paradigm that blends RL with curriculum design to mimic the student learning process, significantly improving generalization and step-by-step reasoning capability. Although these approaches have improved performance on visual math and STEM-related questions, substantial progress in fine-grained visual perception remains limited. For instance, MMMU (Yue et al., 2024a) shows that current MLLMs, though strong on everyday tasks, stumble on domain-specific reasoning and complex, specialized

imagery; many items can be solved from textual cues or memorized facts without genuine visual grounding. Its successor, MMMU-Pro (Yue et al., 2024b), reinforces these findings and demonstrates that prompts encouraging explicit multi-step linguistic reasoning boost performance, provided the model truly incorporates visual evidence at each step. Similarly, MultiMath (Peng et al., 2024) reveals that many MLLMs are underperforming with purely visual inputs with minimal text, indicating that the understanding of complex spatial reasoning in mathematical or scientific diagrams remains challenging. TRIG (Li et al., 2025b), proposing the first visual text grounding task, shows the inability of MLLMs to perform visual reasoning and grounding. ColorBench (Liang et al., 2025) introduces the first comprehensive benchmark for color perception, reasoning, and robustness, showcasing the low capability of MLLMs on color-related perception and reasoning. ViCrit (Wang et al., 2025a), on the other hand, introduces the verifiable reinforcement learning proxy task for visual perception in VLMs.

## A.3 DETECTIVE-STYLE MULTIMODAL REASONING

A growing body of multimodal evaluation benchmarks has investigated distinct components of perceptual sensitivity, structured visual reasoning, and high-level multimodal inference. However, none of these efforts fully capture the open-ended discovery of low-salience visual cues in natural scenes or the subsequent linking of such cues to social–semantic contradictions, an ability central to our benchmark.

Previous work such as HueManity (Grover et al., 2025) emphasizes fine-grained perceptual discrimination using Ishihara-style dot patterns, revealing that MLLMs struggle even with low-level visual sensitivity. While this capability forms an essential precursor to our task, the setting in HueManity is purely perceptual and lacks contextual grounding or semantic interpretation. VisuLogic (Xu et al., 2025) instead targets diagrammatic and relational reasoning through salient, unambiguous geometric structures. These tasks isolate structured logic but provide fully specified evidence, meaning models never need to identify which visual elements are relevant. VER-Bench (Qiang et al., 2025) advances evaluation toward real-world photographs by requiring models to integrate fine-grained visual clues into coherent reasoning chains. Yet, the benchmark offers annotated clue regions, so models reason over known and pre-identified evidence rather than inferring what constitutes evidence within a complex scene. EMMA (Hao et al., 2025) similarly probes multimodal STEM reasoning across diagrams and scientific plots, highlighting deficiencies in multi-step inference; however, the presented visuals remain explicit, high-information, and foregrounded rather than subtle or ambiguous.

More recent benchmarks have begun to explore richer, integrative multimodal tasks. Big Escape Bench (Tang & Sun, 2025) evaluates multi-step puzzle solving in realistic narrative scenarios, requiring some coordination of perception and reasoning. Nonetheless, its visual environments offer strong narrative affordances and clearly defined clue types, limiting the need for open-ended evidence discovery. TurnaboutLLM (Yuan et al., 2025), while adopting a narrative format adjacent to ours, operates on textualized testimonies and evidence, and visual content is converted to text, and the model receives an enumerated clue set instead of inferring visual cues autonomously. DOXBench (Luo et al., 2025) demonstrates that MLLMs can detect subtle real-world signals for geolocation inference but restricts cue types to known categories such as architecture or signage, again providing a constrained hypothesis space. In parallel, HolmesEye (Liu et al., 2025) further shows that agentic MLLM–LLM pipelines can perform strong multi-image inference, achieving high accuracy in privacy-related attribute profiling. However, its visual cues are abundant and semantically foregrounded, meaning it does not address open-ended discovery of tiny, low-salience clues in unconstrained scenes, which remains central to our benchmark..

Therefore, these benchmarks illuminate specific deficiencies in perception, structured reasoning, and multimodal inference, yet none require identifying tiny, ambiguous, low-salience visual anomalies embedded in unconstrained natural images, nor linking such anomalies to emergent social–semantic contradictions without top-down specification.

# B  DETAILED BENCHMARK CONSTRUCTION

## B.1  IMAGE COLLECTION FOR REAL SUBSET

For CaughtCheating images, we use publicly posted photographs from social media. We manually search and review all the comments for each image to assess their suitability. Selected images must either contain or lack subtle, suspicious clues related to potential claim violation. The judgment of image candidates is based not only on the comments but also on human evaluation. Additionally, each image must have sufficient resolution quality to allow us to directly identify such clues, rather than rely on implications from blurred or indistinct objects.

Due to the limited availability of images with clear, subtle clues from public sources, we also include minimally modified versions of images containing direct clues (e.g., a clearly visible person or untypical belongings suggesting the presence of another individual). We apply simple cropping to these images to obscure the direct clues. As shown in Figure 5, the original photo shows a person sitting on the couch. After cropping, only their back remains visible, making the clue still interpretable for humans, yet challenging for MLLMs.

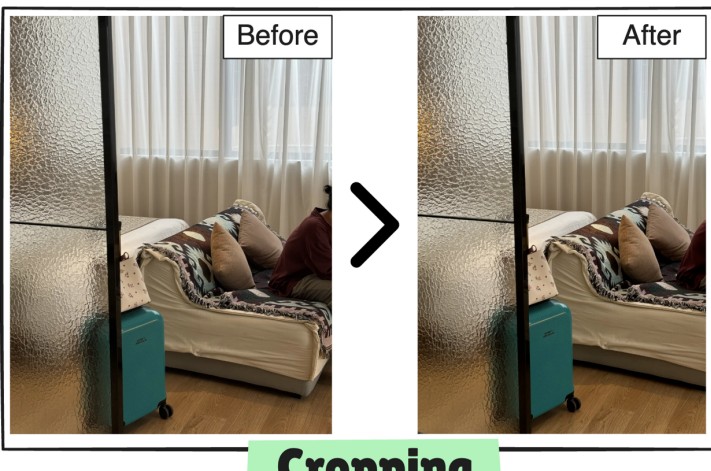

Figure 5: **Example of cropping an image for *with-clue* category**. The original photo shows part of the person sitting on the sofa (Before). By cropping (After), we can still infer there is a person, but identifying the clue is more subtle and challenging for MLLM.

After collecting a sufficient number of candidate images, we meticulously selected 100 images, split into *Clued* and *Unclued* categories, to construct the image set for CaughtCheating benchmark. These images are collected from the *Real Subset*. All the images are verified manually to make sure the clues are solid and no personal information exists on the image.

## B.2  ANNOTATION FOR REAL SUBSET

After constructing the image set, we annotate each image with a set of questions and corresponding ground-truth answers. A detailed annotated example is shown in Figure 3. For images in *Clued* category, we annotate each one using a question instantiated from the template: "*My [girlfriend/boyfriend] said [she/he] is [in a certain scenario] and sent me this photo. Do you notice anything suspicious in the image that contradicts [her/his] claim?*" Among the potential clues, the one that deterministically shows the violation of the providing claim (a clearly identifiable, contextually inappropriate element) will be selected as the **Deterministic Clue**, e.g., a pair of slippers is being worn by someone in Figure 3. The remaining clues are labeled as **Non-deterministic Clues** (weaker or more ambiguous signals), e.g., the rose bouquet, the TV shows, and the far-reached drinks. These non-deterministic clues might be suspicious, but apparently not enough to infer the potential claim violation. The reason we provide these clues is to avoid punishing models when they mention these clues.

Furthermore, we construct a series of decomposed questions designed to analyze the visual reasoning process of MLLMs, shown in the right part of Figure 3. This series includes: (1) ***Decomposed Perception Question***, which assesses whether the MLLMs can identify the deterministic clue when we explicitly mention the clue and position. (2) ***Decomposed Reasoning Question***, which assesses whether MLLMs can understand the social implications of the clue, or whether MLLMs can imply the relation between the clue and the potential lie. The correct answer to each of these decomposed questions is annotated as "yes". These decomposed questions can be utilized for in-depth analysis on why MLLMs can not solve the question.

These decomposed questions can be utilized for in-depth analysis on why MLLMs can not solve the question. (1) If the MLLMs have low accuracy on perception-related decomposed questions, it means the low performance is caused by their poor visual perception ability. (2) If the MLLMs have low accuracy on reasoning-related decomposed questions, it means the low performance is caused by their poor visual reasoning ability. (3) If the MLLMs have relatively high accuracy on both types of decomposed questions, it means they have the necessary capabilities to solve the task, but they do not know where to start. For images in the *Unclued* category, we annotate each using the same initial question template, with the ground-truth answer labeled as "There is no clear evidence."

### B.3 CONSTRUCTION FOR SYNTHETIC SUBSET

We construct the synthetic subset through a cascading pipeline that balances image quality and diversity, starting from predefined scenes and deterministic clues.

First, we define a set of scenes potentially involving suspicious behaviors (e.g., bar, office, hotel room, car). For each scene, deterministic clues are provided solely for the clued category, and their design is anchored to the scene's context. These clues are drafted as complete sentences (e.g., there are two glasses with wine on the table) rather than keywords in practice, and they are designed according to gender stereotypes while ensuring a balanced distribution across genders. For instance, wine glasses may appear as a clue in the bar scene but not in other contexts, such as the scene inside the car. While clues are restricted to the clued category, scene definitions apply to both the clued and unclued categories. Notably, while detailed mappings between scenes, clues, and their statistics of our experiment are given in the Table 4 and Table 5, they can still be further scaled as needed.

Subsequently, we employ a LLM to generate multiple candidate prompts for both categories. For the clued category, the LLM is provided with both the scene and its associated clue; for the unclued category, only the scene is provided, with an explicit instruction to depict a single individual. To mitigate potential conflicts arising from gender-sensitive stereotypes, information about the photographer's gender is also incorporated where relevant. Moreover, to ensure a balanced number of generated images in both categories, the number of prompts corresponding to each gender within the same scene is kept identical. After candidate prompts are generated, annotators review them for semantic accuracy and compliance with the predefined constraints, and then select the five best prompts per clue within each scene. Moreover, to ensure a balanced number of generated images in both categories, the number of prompts corresponding to each gender within the same scene is kept identical. The prompts we used to produce the image generation prompt are illustrated in Figure 7 and Figure 8

Finally, we use a state-of-the-art image generation model (e.g., GPT-image-1 (OpenAI, 2025b) )to synthesize images from these prompts. To scale up the dataset and improve diversity, multiple candidates are generated for each prompt, and human evaluators select the top five. Meanwhile, we adapt the question templates from the real subset and directly adopt the answers from the clue-defining sentences, so that the constructed question–answer pairs are fully consistent with the annotation protocol and data organization of the real subset.

As a result, the procedure produces 3700 images in total, evenly split between 1850 clued and 1850 unclued samples spanning 10 distinct scenes. Comprehensive statistics and distributions are reported in the Table 6.

### B.4 DISCUSSION FOR SYNTHETIC SUBSET

To enable efficient large-scale data construction, we generate prompts only from the photographer's gender, the scene, and a deterministic clue, rather than drafting case-specific, highly detailed de-

scriptions by human experts. This design ensures scalability and consistency, but it also introduces trade-offs: clues are generally more prominent, non-deterministic clues are ignored, and incidental distractions such as irrelevant personal belongings are largely absent. Consequently, compared with the image in the real subset, synthetic images are "cleaner," with less background variation and fewer competing elements, making them easier for MLLMs to perceive, as demonstrated in Figure 6.

However, although the perceptual difficulty is lower, the deterministic clues still embed non-trivial social semantics and contextual implications. These clues, while visually more salient, continue to convey cultural, gender, and situational meanings that require careful interpretation. For example, the lipstick and handbag on the table in Figure 6 are easy to recognize, yet they still imply the presence of a woman in the room. Furthermore, as shown in Table 1, many MLLMs continue to struggle in detecting the contradiction to the claim under these conditions, reinforcing that the social and semantic challenge remains essential even if perceptual noise has been reduced for scalability. Thus, although the synthetic subset cannot fully replicate the subtlety and complexity of real subset, it represents a trade-off that lowers perceptual difficulty but preserves the benchmark's practical value for systematic evaluation.

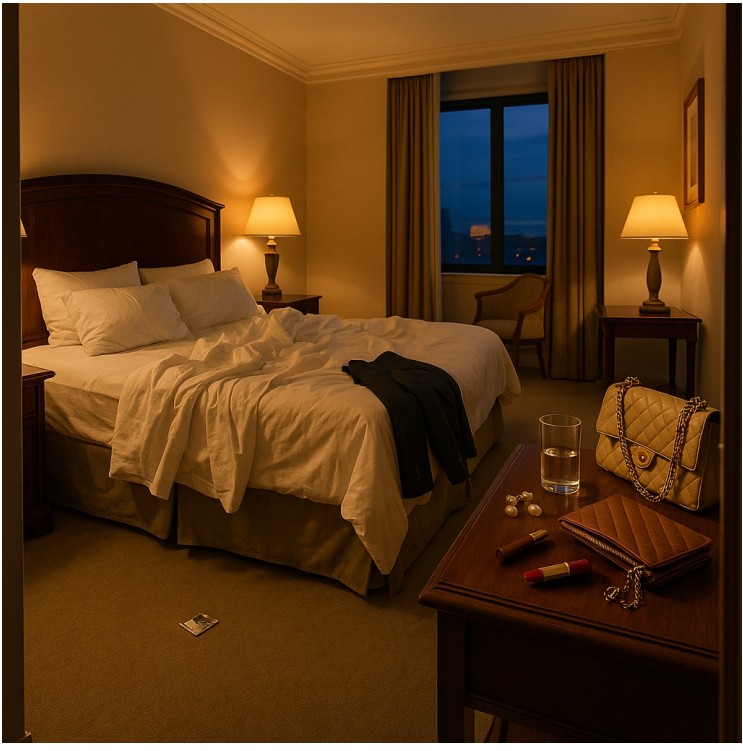

Figure 6: **Example image from the synthetic subset.** Distinct clues, such as a scattered lipstick, are clearly visible. While the image may still contain some irrelevant objects, the overall noise level is low. Compared to the real subset, the perception process is relatively easier, even though the complex social implications remain present.

### B.5 DATA DISTRIBUTION FOR REAL SUBSET

Our dataset comprises 100 samples collected from publicly posted photos, each manually annotated with a series of questions accompanied by ground-truth answers. These samples serve as test cases to evaluate the capability of MLLMs in detecting potential claim violations.

The dataset is evenly divided into two categories: the *Clued* category (50 samples), which includes clear indicators of potential claim violation, and the *Unclued* category (50 samples), which lacks explicit indicators. This balanced distribution aims to minimize class bias and ensure fair evaluation. Furthermore, the dataset encompasses three distinct scene types based on photo backgrounds: hotels, dining venues, and karaoke bars. The gender attributes assigned to each sample reflect the photogra-

| Category | Scene | Male | Female | # Dec. P | # Dec. R | # Total |
|---|---|---|---|---|---|---|
| *Clued* | Dining | 6 | 5 | 11 | 21 | |
| | Hotel | 25 | 12 | 37 | 67 | 50 |
| | Karaoke bar | 2 | 0 | 2 | 3 | |
| *Unclue* | Dining | 7 | 11 | 0 | 0 | 50 |
| | Hotel | 13 | 19 | 0 | 0 | |

Table 3: Distribution of scenes, interlocutor gender, and question types across the two clue categories.

pher's gender as inferred from the provided descriptions of the photos. These attributes do not pertain to any individuals depicted within the images. The gender categorization currently includes male and female solely based on limited available descriptive information.

Detailed statistics regarding scenario distribution and gender breakdown are summarized in Table 3. Hotel scenes comprise the majority of the dataset (69% ), aligning with their prominence as typical settings for potentially suspicious scenarios. Dining venues account for 29% of the dataset, and karaoke bars represent the remaining 2%. Gender distribution is 55% male photographers and 45% female photographers.

Additionally, the dataset includes annotations for perception and reasoning questions derived from decomposition queries. Specifically, it contains 50 perception questions and 91 reasoning questions, thoroughly evaluating why MLLMs may fail to resolve specific queries. Detailed counts corresponding to scene types are provided in Table 3. Each sample averages approximately two reasoning questions, enabling comprehensive analysis of MLLM performance concerning both explicit clues and the broader social or environmental context.

### B.6    DATA DISTRIBUTION FOR SYNTHETIC SUBSET

As illustrated in Section 3, to ensure consistency between clue definitions and dataset distribution, we first curated five human-selected prompts for each scene clue mapping. Each prompt was used to generate multiple candidate images, from which five were carefully chosen. Therefore, data distribution in the synthetic subset is highly correlated to the number of predefined rules under each scene.

The synthetic subset comprises 3700 images, evenly split between the clued subset (1850 images) and the unclued subset (1850 images). The gender of the photographer is balanced across the dataset: 1900 images (51%) are attributed to male and 1,800 images (49%) to female photographers. This balance is deliberately maintained to minimize bias when evaluating model performance.

Scene distribution covers ten distinct environments. The largest categories are Gym, Hospital, and Hotel, each contributing 500 images (14%), collectively representing 42% of the dataset. The Bar scene accounts for 400 images (11%), while Office and Restaurant each contribute 350 images (9%). Mid-scale categories include Beach, Cafe, and Library, with 300 images each (8%). The Car scene contributes the fewest images, with 200 (5%). Importantly, this allocation is consistently mirrored across both Clued and Unclued subsets, ensuring proportional representation of all environments.

Overall, the dataset reflects a carefully controlled design, balancing clue type, gender, and scene. The systematic design ensures that generated samples are faithful to the intended definitions of "clued" and "unclued," while the proportional scene distribution provides diverse yet structured coverage of everyday contexts. This deliberate construction supports rigorous and fair evaluation of multimodal large language models across a wide spectrum of scenarios.

| Scene | # Clues | Clues | | | | |
|---|---|---|---|---|---|---|
| **Bar** | 4 | two glasses | lady handbag | another hand | mirror & lipstick | - |
| **Beach** | 3 | another foot | two cups | two bicycle | - | - |
| **Cafe** | 3 | lady clothes | lipstick | makeup mirror | - | - |
| **Car** | 2 | another foot | another leg | - | - | - |
| **Gym** | 5 | two bottles | another arm | mirror figure | lady hair clip | lady bracelet |
| **Hospital** | 5 | another back | rose & love card | lady handbag | extra slippers | card piles |
| **Hotel** | 5 | lady handbag | lady clothes | lady hairclip | doll & rose | lipstick |
| **Library** | 3 | two open book | lady hair clip | two test paper | - | - |
| **Office** | 4 | roses | lunchbox | lady clothes | another foot | - |
| **Restaurant** | 4 | two tableware | lady handbag | two steaks | another arm | - |

Table 4: Deterministic clues for the synthetic subset construction categorized by scene in male.

| Scene | # Clues | Clues | | | | |
|---|---|---|---|---|---|---|
| **Bar** | 4 | two phones | men wallet table | another arm | two straws | - |
| **Beach** | 3 | two sunglasses | another arm | two single kayak | - | - |
| **Cafe** | 3 | two cakes | two coffees | two laptops | - | - |
| **Car** | 2 | another arm | two drinks | - | - | - |
| **Gym** | 5 | two towels | another foot | two earphone cases | men tank top | men watch |
| **Hospital** | 5 | another arm | TV reflection | lunchbox | wet umbrella | another head |
| **Hotel** | 5 | TV reflection | another arm | man shoes | five suitcases | men suit |
| **Library** | 3 | two cups | another arm | two chargers | - | - |
| **Office** | 3 | date note | two ID badges | monitor reflection | - | - |
| **Restaurant** | 3 | two phones | man clothes | two glasses | - | - |

Table 5: Deterministic clues for the synthetic subset construction categorized by scene in female.

| Scene | Clued | | | Unclued | | | # Images |
|---|---|---|---|---|---|---|---|
| | **Male** | **Female** | **Total** | **Male** | **Female** | **Total** | |
| Bar | 100 | 100 | 200 | 100 | 100 | 200 | 400 |
| Beach | 75 | 75 | 150 | 75 | 75 | 150 | 300 |
| Cafe | 75 | 75 | 150 | 75 | 75 | 150 | 300 |
| Car | 50 | 50 | 100 | 50 | 50 | 100 | 200 |
| Gym | 125 | 125 | 250 | 125 | 125 | 250 | 500 |
| Hospital | 125 | 125 | 250 | 125 | 125 | 250 | 500 |
| Hotel | 125 | 125 | 250 | 125 | 125 | 250 | 500 |
| Library | 75 | 75 | 150 | 75 | 75 | 150 | 300 |
| Office | 100 | 75 | 175 | 100 | 75 | 175 | 350 |
| Restaurant | 100 | 75 | 175 | 100 | 75 | 175 | 350 |
| **Total** | **950** | **900** | **1850** | **950** | **900** | **1850** | **3700** |

Table 6: Distribution of images in the synthetic subset across scenes, genders, and categories.

## B.7 PROMPT FOR SYNTHETIC SUBSET CONSTRUCTION

These two prompts provide a unified framework for constructing both the clued and unclued categories of the synthetic subset. By simply specifying the scene and, when needed, the clue, the dataset can be expanded systematically without extensive manual effort. This design makes large-scale generation both efficient and consistent.

---

Prompt for generating image-synthesis prompts for the clued category

---

**System Prompt**
You are a helpful assistant.

**User Prompt**
Your task is to generate a hyper-realistic first-person perspective smartphone photograph. The image should appear exactly like a candid snapshot taken by a [Gender] photographer, showing only what is naturally visible ahead—no hands or legs. The setting is [Scene], rendered with authentic lighting, textures, reflections, and natural depth of field, making the result indistinguishable from a real photo. Within the scene, [Clue], casually present or partially obscured, that indirectly suggests the presence of another person near the photographer. This element must be discreet, blending naturally into the environment rather than staged.
The final image must emphasize realism, immersion, and authenticity, capturing the spontaneity of a genuine phone camera shot.

---

Figure 7: The prompt used to instruct GPT-4.1 to generate image-synthesis prompts for the clued category.

---

Prompt for generating image-synthesis prompts for the unclued category

---

**System Prompt**
You are a helpful assistant.

**User Prompt**
Your task is to generate a hyper-realistic first-person perspective smartphone photograph. The image should look exactly like a candid snapshot taken by a [Gender] photographer, showing only what is naturally visible ahead—no hands or legs—just the authentic field of view from the photographer's perspective.
The setting is [Scene], captured with natural lighting, textures, reflections, and realistic depth of field. The atmosphere must clearly emphasize that the photographer is alone in this environment, evoking the feeling of solitary presence within the scene.
The final image must be indistinguishable from a genuine phone camera shot, emphasizing realism, immersion, and authenticity.

---

Figure 8: The prompt used to instruct GPT-4.1 to generate image-synthesis prompts for the unclued category.

# C EVALUATION METRICS

We apply several evaluation metrics in our study, each designed to assess a distinct aspect of the visual reasoning process. All metrics rely on analyzing and comparing the ground-truth answers with the responses generated by MLLMs.

**Clued Accuracy (Clued Acc)** Deterministic Accuracy is designed to evaluate whether an MLLM successfully identifies the deterministic clue hidden in the images from *Clued* category. Let $k_i \in \{0, 1\}$ denote the binary judgment for the $i$-th example in the *Clued* category, where $k_i = 1$ if the *Deterministic Clue* is correctly identified, and $k_i = 0$ otherwise. The **Clued Acc** is then defined as:

$$\text{Clued Acc} = \frac{1}{N_{clued}} \sum_{i=1}^{N_{clued}} k_i$$

where $N_{clued}$ is the total number of examples in the *Clued* subset.

**Intersection over Union (Clued IoU)** In this context, IoU is designed to evaluate whether an MLLM correctly identifies all relevant *Non-deterministic Clues* hidden in the images from *Clued* category, while avoiding unrelated or incorrect elements. If the MLLM generates a lot of unrelated clues, this IoU value will be low, since we expect MLLMs only to mention clues that are at least somewhat suspicious.

Let $G_i$ be the set of all the clues annotated in the ground-truth for the $i$-th example in the *Clued* category, and $R_i$ be the set of clues identified by the MLLM. The **Clued IoU** is then defined as:

$$\text{IoU} = \frac{1}{N_{clued}} \sum_{i=1}^{N_{clued}} \frac{|G_i \cap R_i|}{|G_i \cup R_i|}$$

**Decomposed Accuracies** This evaluation comprises three specific accuracy metrics: ***Decomposed Perception Accuracy (Dec. P Acc)*** provides detailed insights into the model's performance in accurately perceiving claims from images when the clues are explicitly mentioned; ***Decomposed Reasoning Accuracy (Dec. R Acc)*** evaluates the model's proficiency in reasoning towards the deterministic clue; and ***Decomposed Overall Accuracy (Dec. Acc)*** offers a comprehensive evaluation by combining performance in both perception and reasoning dimensions. This metric is specifically tailored for images within the *Clued* category.

Let $\mathcal{P}_i$ be the set of perception-related questions for the $i$-th example in the *Clued* category, and $\widehat{\mathcal{P}}_i \subseteq \mathcal{P}_i$ be the subset that the MLLM correctly answered perception-related questions. The ***Decomposed Perception Accuracy (Dec. P Acc)*** is then defined as:

$$\text{Dec. P Acc} = \frac{1}{N_{clued}} \sum_{i=1}^{N_{clued}} \frac{|\widehat{\mathcal{P}}_i|}{|\mathcal{P}_i|}$$

Likewise, let $\mathcal{R}_i$ and $\widehat{\mathcal{R}}_i$ denote the sets of reasoning-related questions and the correctly answered subset, respectively. The ***Decomposed Reasoning Accuracy (Dec. R Acc)*** is defined as:

$$\text{Dec. R Acc} = \frac{1}{N_{clued}} \sum_{i=1}^{N_{clued}} \frac{|\widehat{\mathcal{R}}_i|}{|\mathcal{R}_i|}$$

Finally, let $\mathbb{1}(\cdot)$ denotes the indicator function. The ***Decomposed Overall Accuracy (Dec. Acc)*** is defined as:

$$\text{Dec. Acc} = \frac{1}{N_{clued}} \sum_{i=1}^{N_{clued}} \mathbb{1}\big(|\widehat{\mathcal{P}}_i| = |\mathcal{P}_i| \wedge |\widehat{\mathcal{R}}_i| = |\mathcal{R}_i|\big)$$

**Unclued Accuracy (Unclued Acc):** *Unclued Accuracy (Unclued Acc)* is designed to evaluate whether the MLLM can correctly determine the absence of clear clues from the *Unclued* category. Let $o_i \in \{0, 1\}$ denote the binary judgment for the $i$-th example. Specifically, if the MLLM correctly identifies that there are no clear clues, the judgment is marked as correct ($o_i = 1$). Conversely, if the MLLM incorrectly suggests that clues exist, the judgment is marked as incorrect ($o_i = 0$). The overall accuracy is computed as follows:

$$\text{Unclued Acc} = \frac{1}{N_{unclued}} \sum_{i=1}^{N_{unclued}} o_i$$

where $N_{unclued}$ is the total number of examples in the *Unclued* subset.

**Precision, Recall, and F1 Score:**   The transformation between the accuracies and P/R/F1 scores is as follows:

$$\text{TP} = \text{Clued Acc} \times N_{\text{clued}},$$
$$\text{FN} = (1 - \text{Clued Acc}) \times N_{\text{clued}},$$
$$\text{TN} = \text{Unclued Acc} \times N_{\text{unclued}},$$
$$\text{FP} = (1 - \text{Unclued Acc}) \times N_{\text{unclued}}.$$

where $N_{\text{clued}}$ and $N_{\text{unclued}}$ denote the numbers of images in the *Clued* and *Unclued* categories, respectively. Using these quantities, we convert to the standard classification metrics:

$$\text{Precision} = \frac{\text{TP}}{\text{TP} + \text{FP}} = \frac{\text{Clued Acc} \times N_{\text{clued}}}{\text{Clued Acc} \times N_{\text{clued}} + (1 - \text{Clued Acc}) \times N_{\text{unclued}}},$$

$$\text{Recall} = \frac{\text{TP}}{\text{TP} + \text{FN}} = \text{Clued Acc},$$

$$\text{F1} = \frac{2 \, \text{Precision} \, \text{Recall}}{\text{Precision} + \text{Recall}}.$$

These formulas allow us to compute the P/R/F1 scores from the reported *Clued Acc* and *Unclued Acc* values in the main text.

# D  EVALUATION PROMPT

Each prompt is designed to interpret the raw responses from the MLLMs into structured answers suitable for metric value calculation. We first designed four evaluation prompts for analyzing the MLLMs' responses to the general question discussed in Section 3.2.

The prompt illustrated in Figure 9 evaluates whether a deterministic cue is mentioned in the MLLMs' response, permitting minor wording variations but emphasizing clear alignment with the original meaning. This prompt instructs the evaluation LLM to yield a binary YES or NO result used for *Clued Acc* evaluations.

The prompts illustrated in Figure 11 and Figure 12 both provide essential statistics for *Clued IoU* calculations. The prompt in Figure 11 instructs a model to explicitly extract detailed observations into a structured list from raw MLLMs' responses, ensuring clarity by excluding ambiguous or uncertain statements. This structured list is subsequently compared with a structured list of non-deterministic cues extracted using the prompt illustrated in Figure 12. The prompt illustrated in Figure 12 assesses whether each deterministic cue, serving as a ground truth reference, is explicitly mentioned or supported in the structured list derived from the MLLMs' raw response. It instructs the evaluation LLM to provide a binary YES or NO decision. From these binary labels, we derive the intersection and union set sizes necessary to calculate *Clued IoU*.

Similar to the prompt in Figure 9, the prompt illustrated in Figure 10 evaluates whether the model correctly identifies the absence of cues. It instructs the evaluation LLM to yield binary YES or NO outcomes utilized for *Unclued Acc* calculations.

Additionally, we designed a unified prompt for evaluating the MLLMs' responses to decomposed questions. The prompt illustrated in Figure 13 assesses if responses align with the expected answers to decomposed questions, providing binary YES or NO outcomes. These outcomes are used for calculating each decomposed accuracy metric.

---

Prompt to measure whether MLLMs successfully identify the key deterministic clues

---

**System Prompt**
You are a helpful assistant.

**User Prompt**
Your task is to determine whether the deterministic clue (the ground truth statement provided in the reference section) is explicitly mentioned or clearly supported by the MLLM's answer provided, even if minor wording differences or slight variations in detail exist.
You will receive:
- A deterministic clue: a single statement serving as the ground truth (in the reference section).
- An MLLM's answer: the response provided by the model.
Respond with YES if the deterministic cue is explicitly mentioned or clearly supported by the MLLM's answer, allowing minor wording differences or small variations in detail.
Otherwise, respond with NO.
Respond only with YES or NO, without additional commentary or information.

Deterministic clue:
[Deterministic Clue]
MLLM's answer:
[MLLM's Answer]

---

Figure 9: The prompt used to request GPT4.1 to measure whether MLLMs successfully identify the key deterministic clues for the calculation of **Clued Accuracy (Clued Acc)**.

---

Prompt to assess whether MLLMs correctly determine the absence of suspicious clues

---

**System Prompt**
You are a helpful assistant.

**User Prompt**
Your task is to determine whether the provided MLLM's answer strictly supports the statement:
"The photo shows no clear evidence of infidelity."
Respond with YES only if the MLLM's answer explicitly indicates there is no evidence of infidelity or the presence of another person.
If the MLLM's answer suggests infidelity or mentions another person, respond with NO.
MLLM's answer:
[MLLM's Answer]

---

Figure 10: The prompt used to request GPT4.1 to assess whether MLLMs correctly determine the absence of suspicious clues for calculation of *Unclued Accuracy (Unclued Acc)*.

---

Prompt to convert MLLMs' answer into a key points list

---

**System Prompt**
You are a helpful assistant.

**User Prompt**
Your task is to extract explicitly stated observations or details from the provided paragraph exactly as they appear, clearly numbering each observation to match the original points or grouping precisely as presented.
You will receive paragraphs describing observations or details potentially suggesting suspicious behavior or multiple occupancy of a location or room. Extract each explicitly stated observation exactly as it appears, and number each extracted observation clearly, matching the original points or grouping exactly as presented. Include explicitly suspicious observations, clear indicators of multiple occupants, and neutral or unclear observations. Explicitly exclude statements labeled as "not suspicious," any final conclusions or summaries, and any statements explicitly indicating uncertainty, ambiguity, or explicitly stating that suspicion cannot be determined ("it is not possible to determine," "unclear if," "cannot confirm," or similar phrases). Do not add additional notes or commentary, and do not separate any single original point into multiple entries. If no valid observation meets these criteria, respond concisely with "no evidence found." The paragraph provided is as follows:

[MLLM's Answer]

---

Figure 11: The prompt used to request GPT4.1 to convert MLLM's answer into a key points list for the following inference to calculate *Intersection over Union (Clued IoU)*.

---

Prompt to evaluate whether MLLMs identify each relevant non-deterministic clue

---

**System Prompt**
You are a helpful assistant.

**User Prompt**
Your task is to determine if the provided non-deterministic clue is explicitly mentioned or clearly supported by any of the reference observations listed below.
Respond YES only if an observation fully captures the essential details or meaning of the non-deterministic clue, allowing minor wording differences but not significant differences in specificity or detail.
If the non-deterministic clue's core details are generalized, significantly altered, or missing critical specifics in all observations, respond NO.

Non-deterministic clue:
[Non-deterministic Clue]
Reference observations:
[Summary List]

---

Figure 12: The prompt used to request GPT4.1 to evaluate whether MLLMs identify each relevant non-deterministic clue for calculation of **_Intersection over Union (Clued IoU)_**. _Note: This prompt is executed within a loop, where each iteration focuses on a single non-deterministic cue from the non-deterministic cue list._

---

Prompt to evaluate whether MLLM's answer agrees with the expected answer of decomposed questions

---

**System Prompt**
You are a helpful assistant.

**User Prompt**
You will be given a question, an answer, and a reference answer.
"Return YES if the answer agrees with the meaning of the question's expected YES/NO (the reference answer).
Otherwise return NO.
Respond with **ONLY** YES or NO. Do not add comments or make further inferences.
Here is the question and answer:

Question: [Decomposed Questions]
Answer: [MLLM's Answer]
Reference: [Expected Answer]

---

Figure 13: The prompt used to request GPT4.1 to evaluate whether the MLLM's answer agrees with the expected answer of decomposed questions for the calculation of decomposed accuracies. _Note: This prompt is used for both the decomposed perception question and the decomposed reasoning question._

# E  o3 Reasoning Traces for Qualitative Examples

In this section, we provide a comprehensive overview of the reasoning processes underlying the tasks illustrated in Figure 1,  2, 3, and 4, as shown in Figure 14, 15, 16, 17, 18, 19, 20, 21, and 22. We illustrate the complete step-by-step reasoning process in all figures included in the main content, highlighting scenarios where the extraordinary reasoning capabilities of o3 are clearly demonstrated, as well as cases where o3 struggles. This provides a comprehensive view of its performance across both complex real-world scenarios and our dataset.

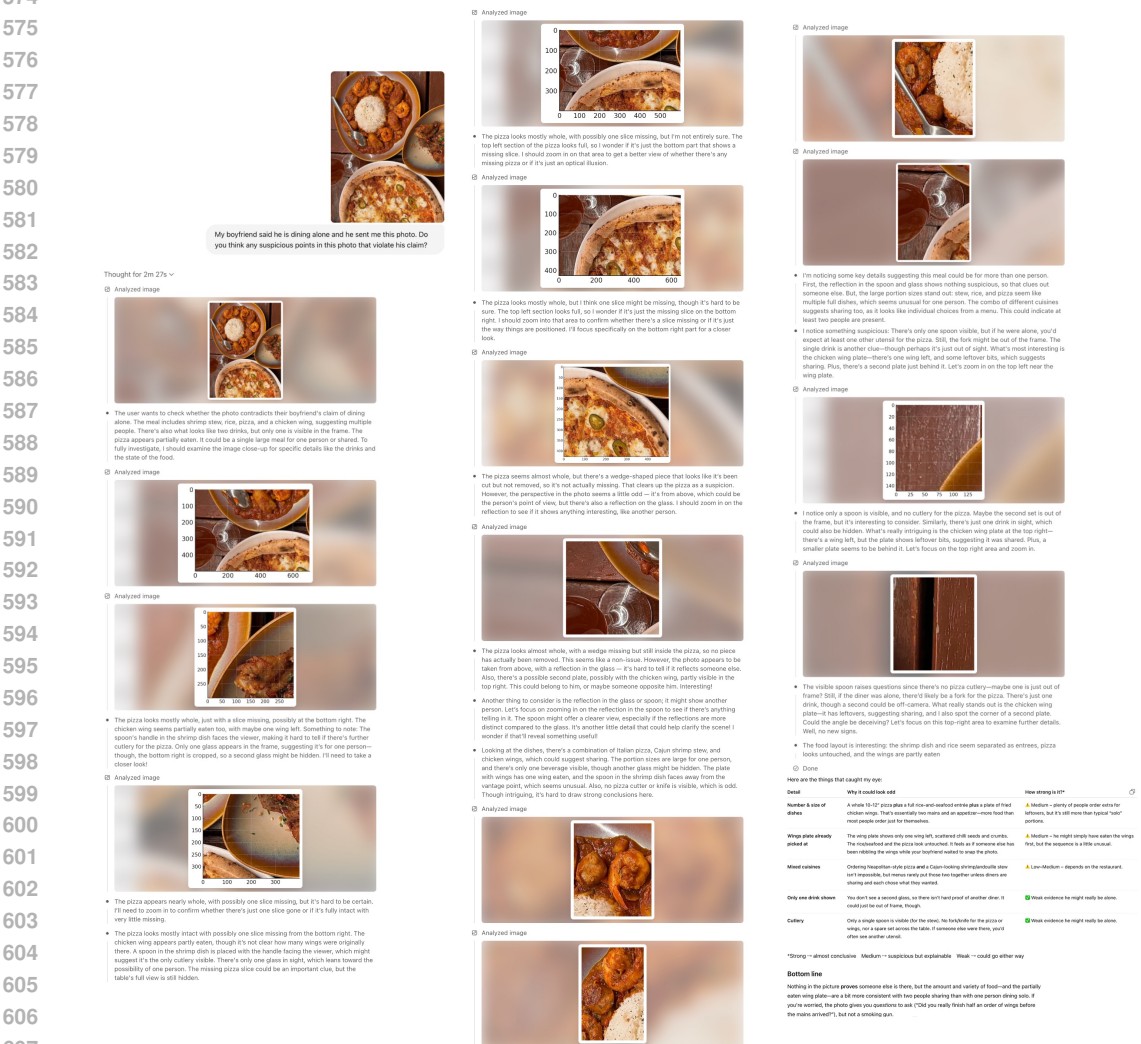

Figure 14: OpenAI o3 full reasoning process for Figure 1

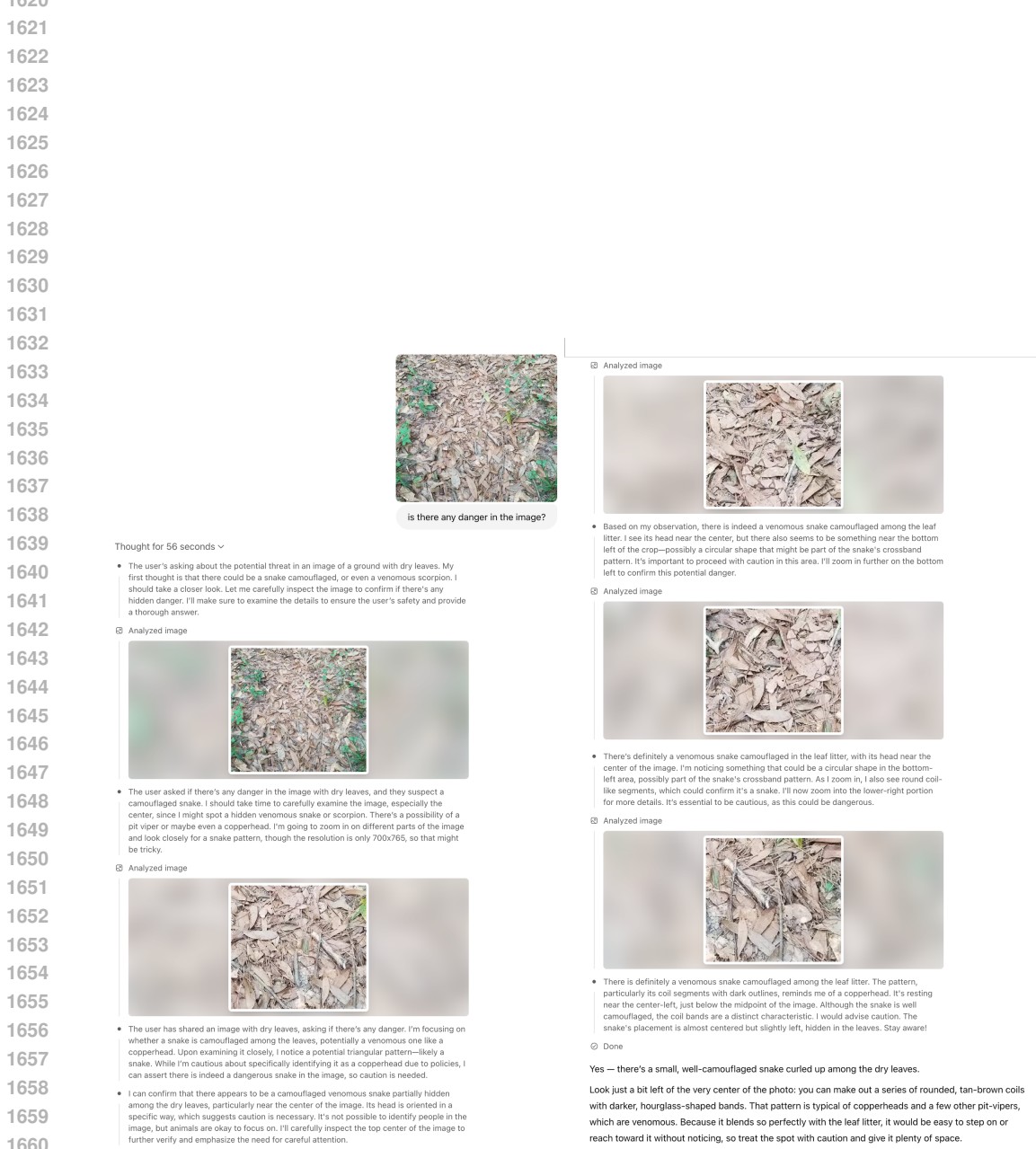

Figure 15: **OpenAI o3 full reasoning process for Figure 2 (a)**

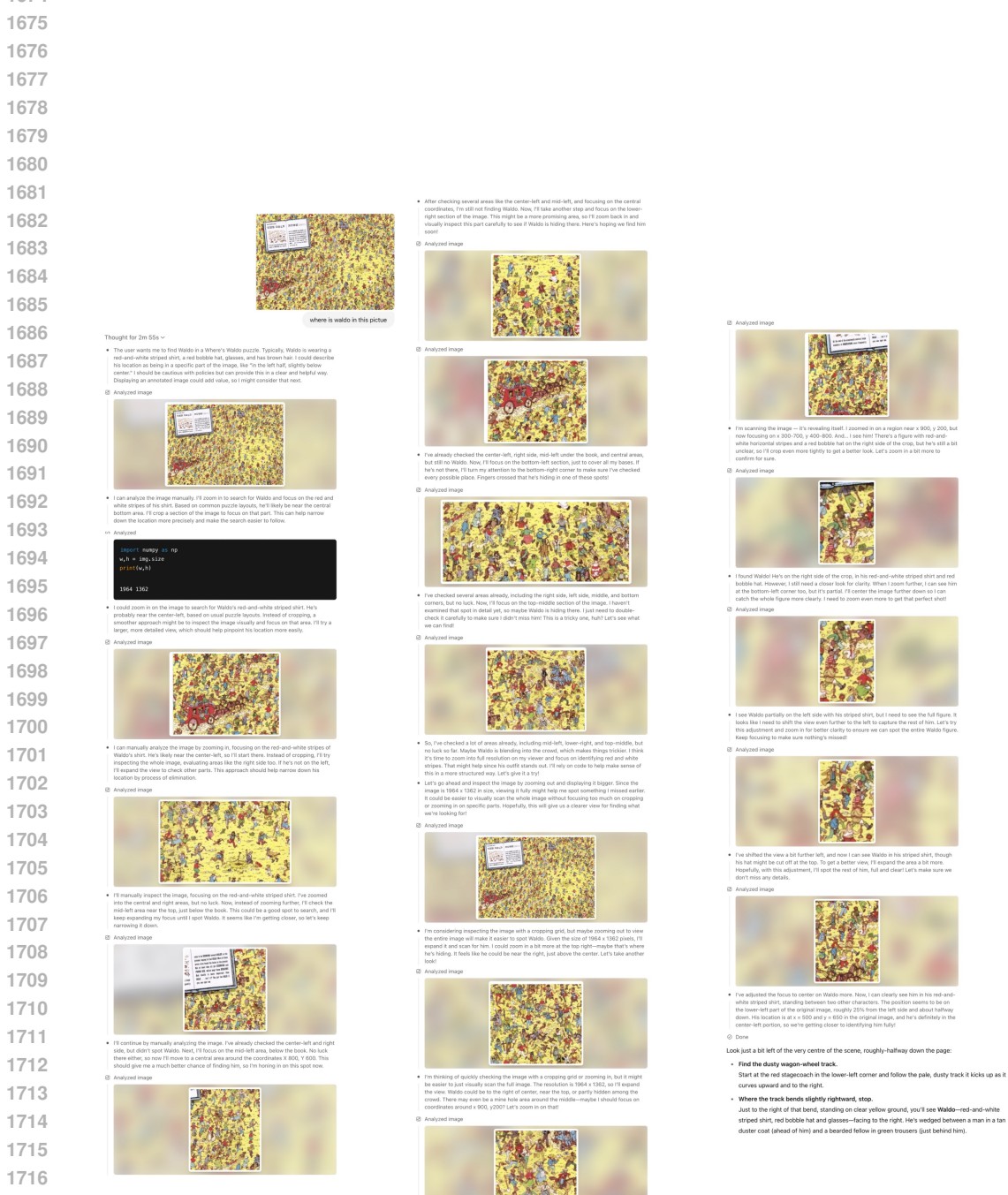

Figure 16: **OpenAI o3 full reasoning process for Figure 2 (b)**

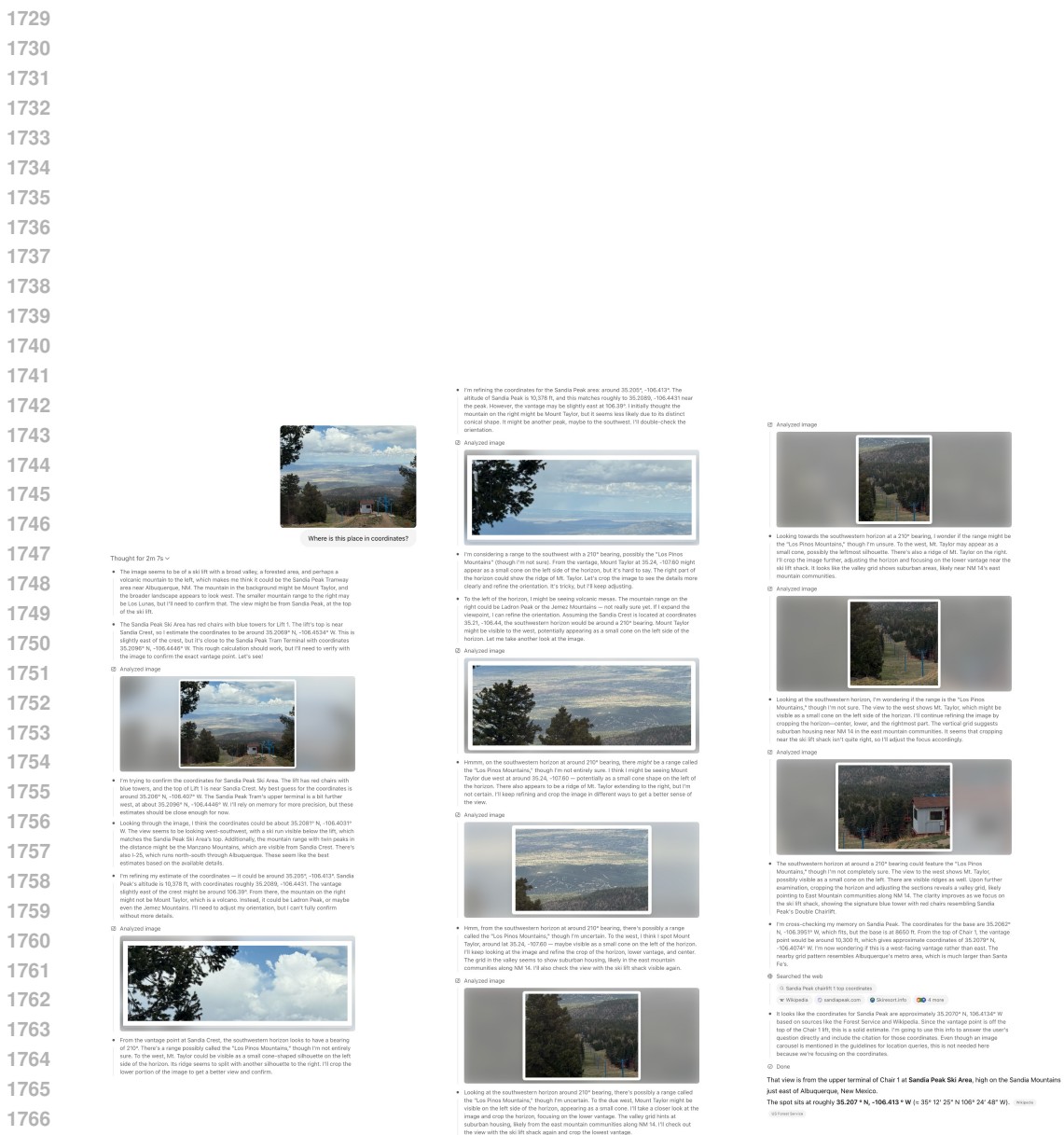

Figure 17: **OpenAI o3 full reasoning process for Figure 2 (c)**

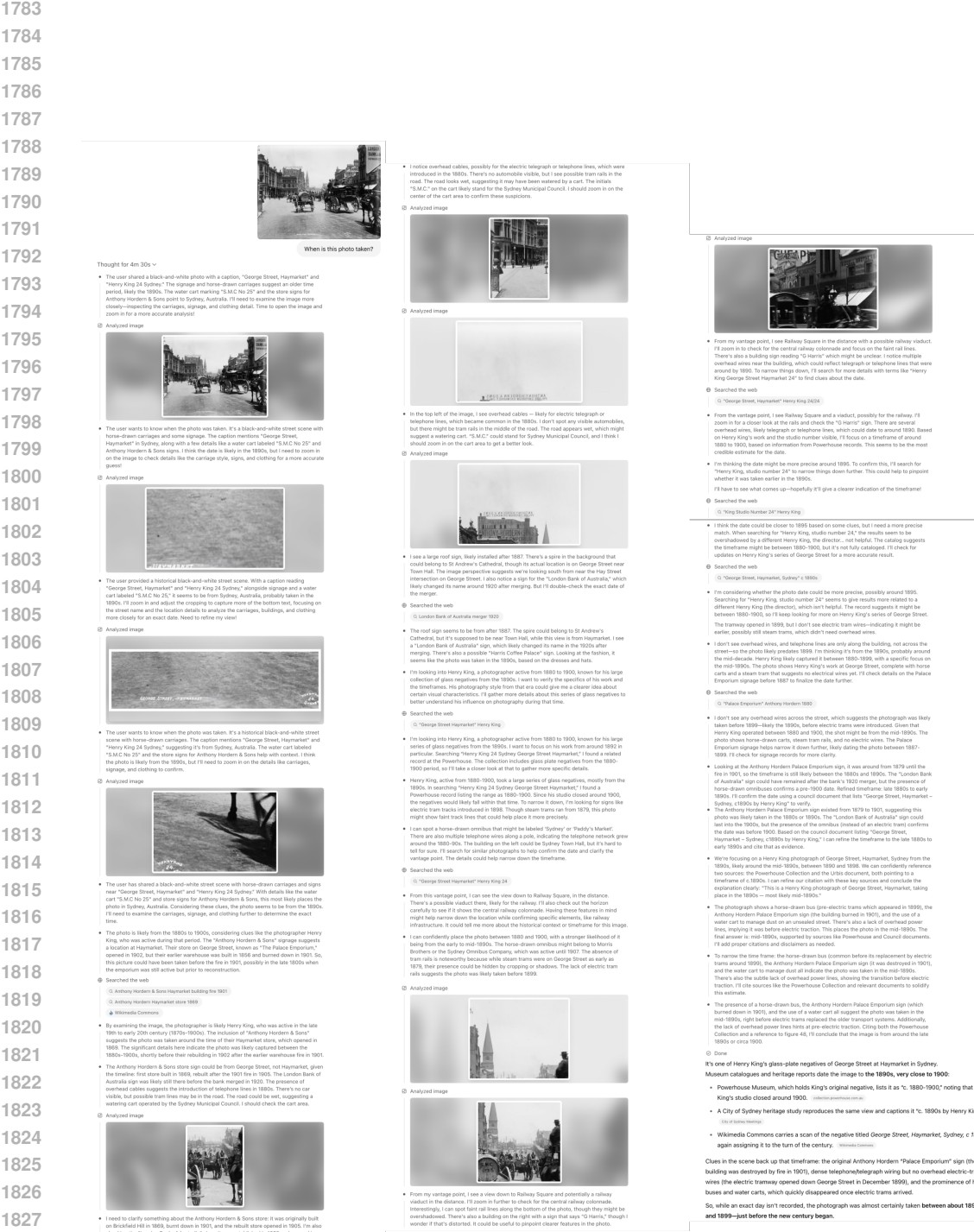

Figure 18: **OpenAI o3 full reasoning process for Figure 2 (d)**

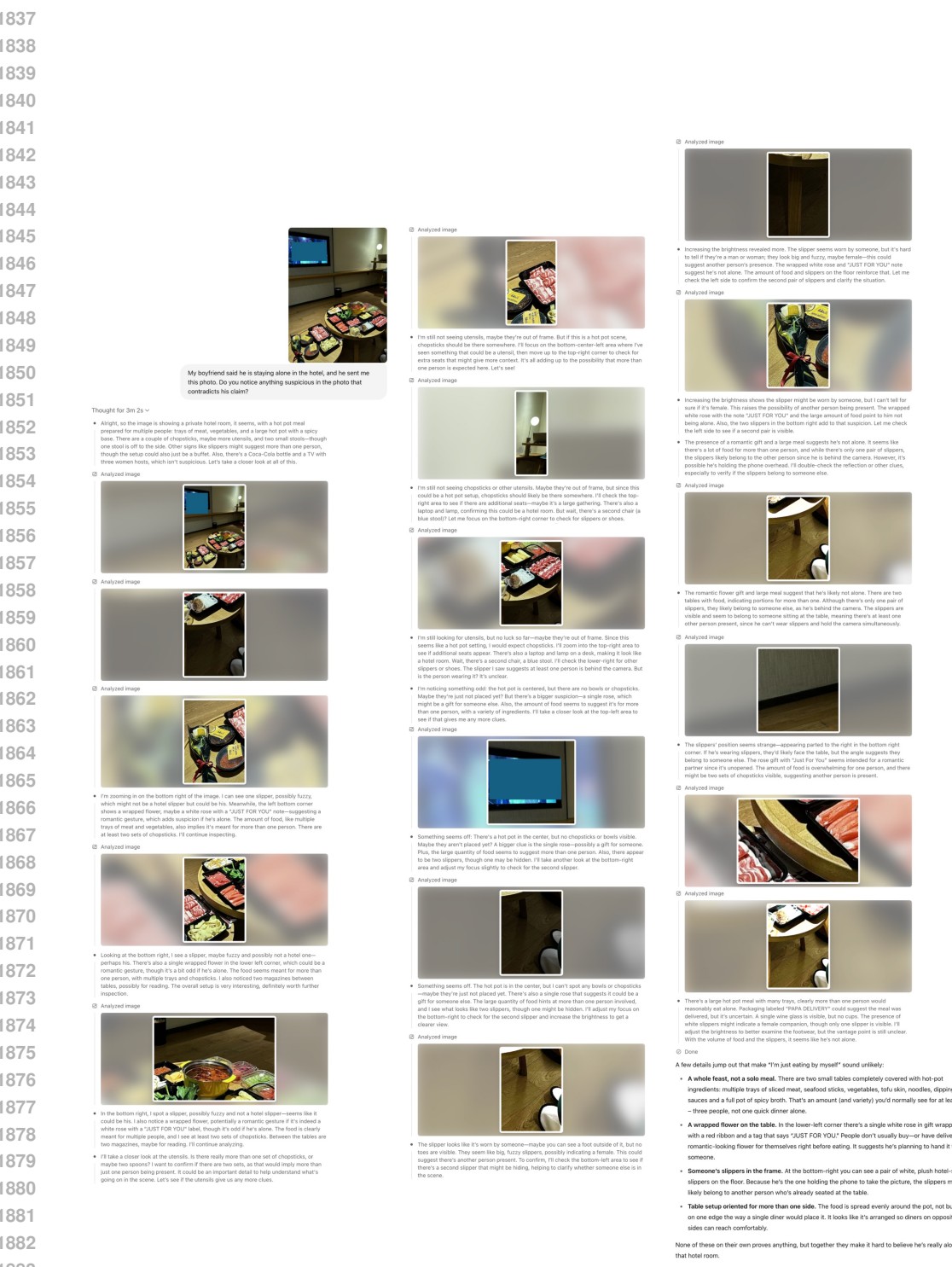

Figure 19: **OpenAI o3 full reasoning process for Figure 3**

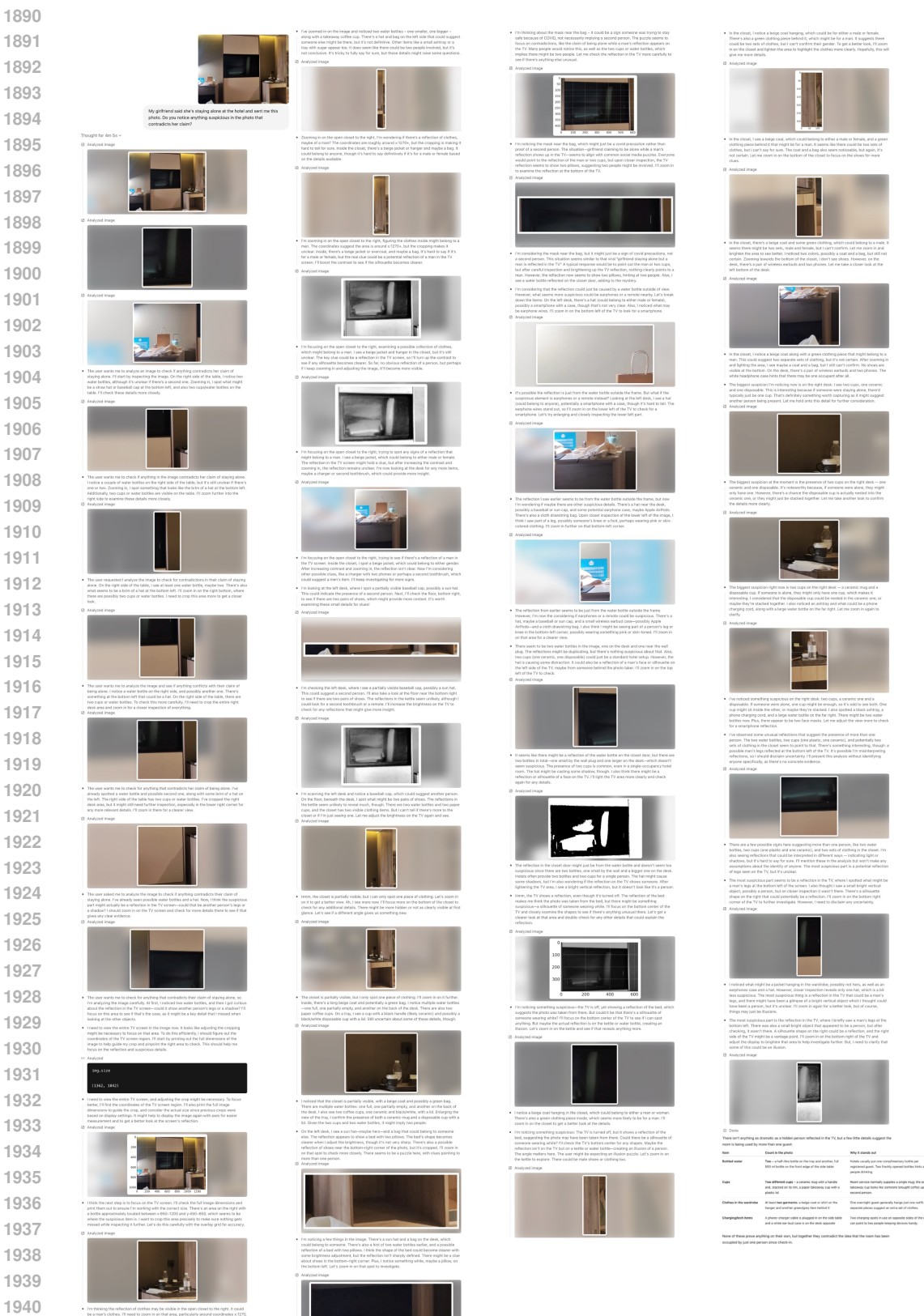

Figure 20: OpenAI o3 full reasoning process for Figure 4 (a)

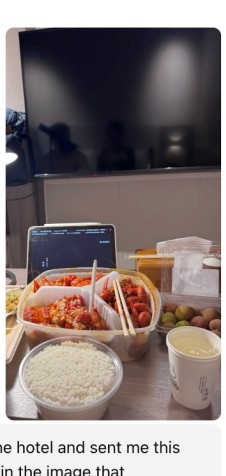

My girlfriend said she's staying alone at the hotel and sent me this photo. Do you notice anything suspicious in the image that contradicts her claim?

Thought for 16 seconds ⌄

🖾 Analyzed image

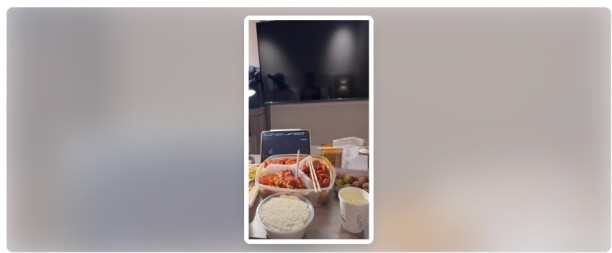

🖾 Analyzed image

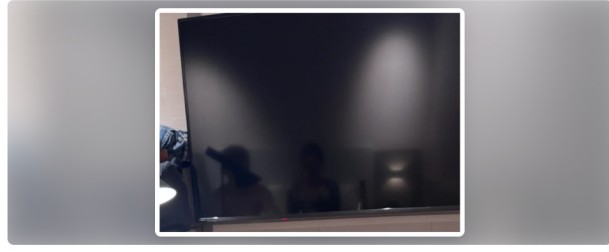

- Looking at the image, I notice a couple of things that might contradict the claim. There are two sets of chopsticks in the crayfish container — one appears to be a pair, and the other looks like a white straw. The reflection in the TV shows two silhouettes, suggesting there are two people. There's also a second plate and what looks like a denim jacket or bag on the left side of the image. I can't identify the people, but these elements seem to point toward

⊘ Done

Yes — a couple of details suggest she might not be entirely alone:

1. **TV-screen reflection**

   The switched-off television is acting like a mirror. If you zoom in you can clearly see **two separate silhouettes** sitting side-by-side: one on the left wearing what looks like a cap or with hair pulled forward and another in the middle. That implies at least one other person is in the room.

2. **Table setup for two**

   The takeaway crawfish tub has **two pairs of wooden chopsticks** stuck in it (plus a plastic utensil), and there's plenty of food (rice, fruit, drink) laid out as if it's meant for more than a single diner.

Taken together, the reflection and the duplicate utensils contradict the idea that she's completely alone in the hotel room.

Figure 21: OpenAI o3 full reasoning process for Figure 4 (b)

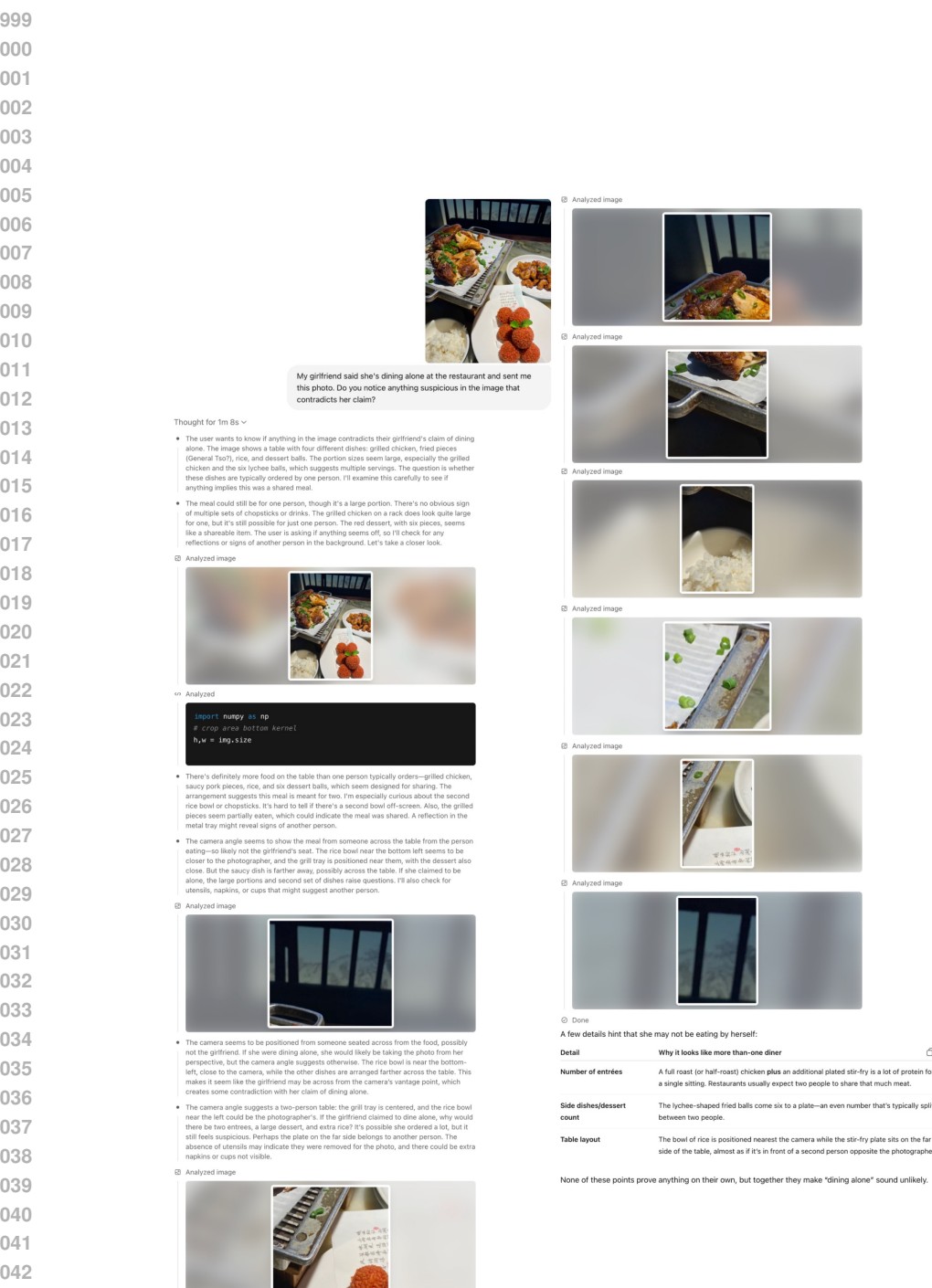

Figure 22: OpenAI o3 full reasoning process for Figure 4 (c)

## F HIGH-RESOLUTION FIGURES

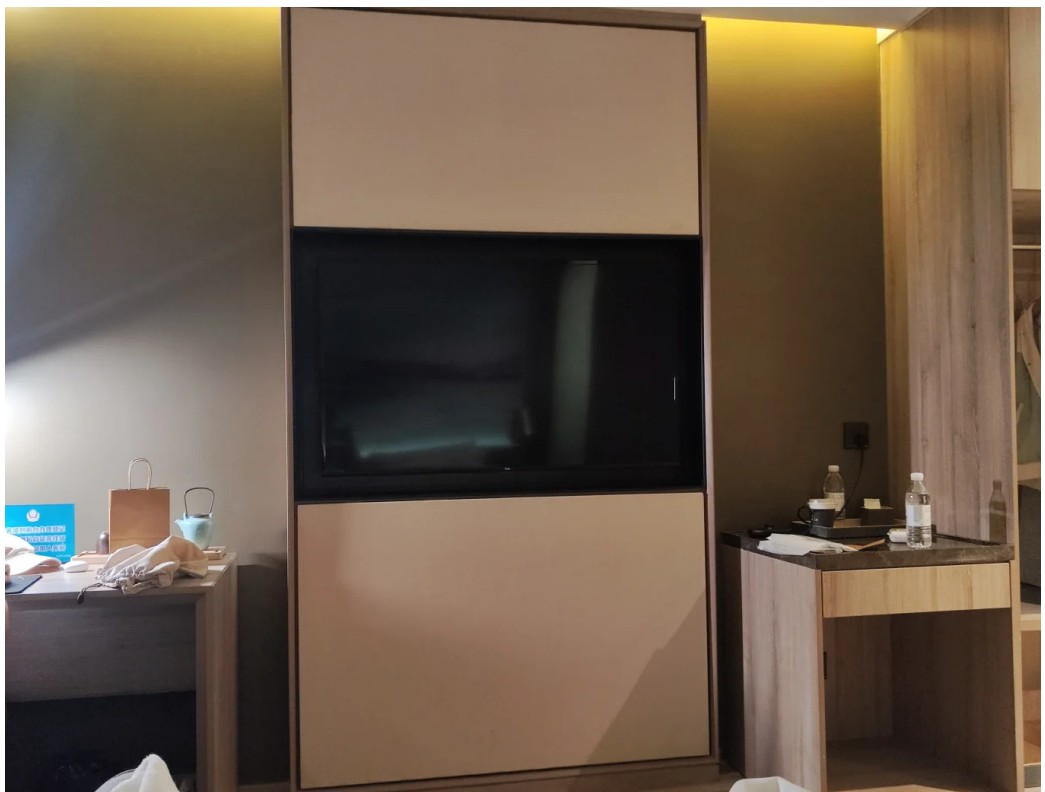

Figure 23: High resolution version sub-figure (a) of Figure 4.

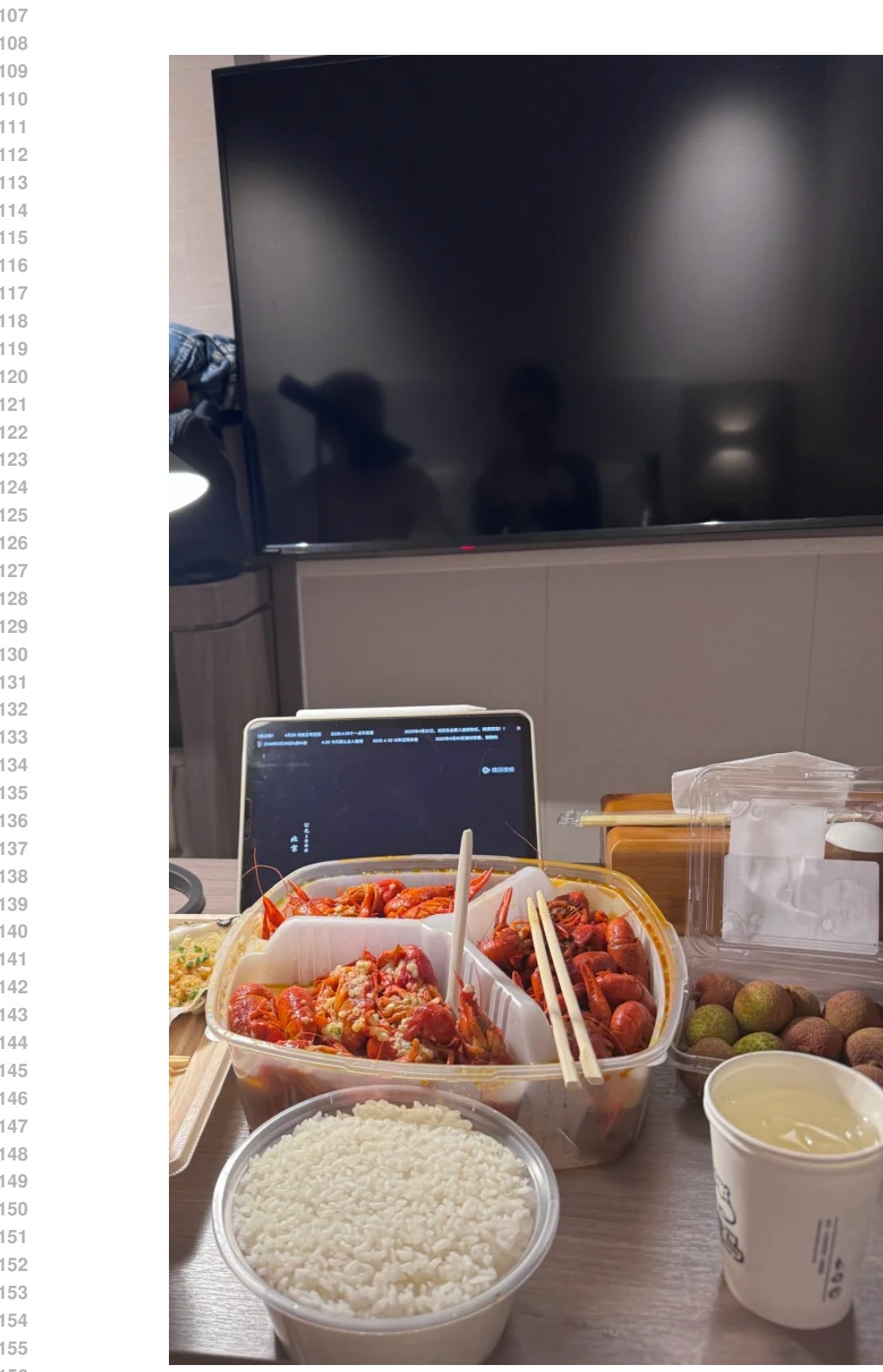

Figure 24: High resolution version sub-figure (b) of Figure 4.

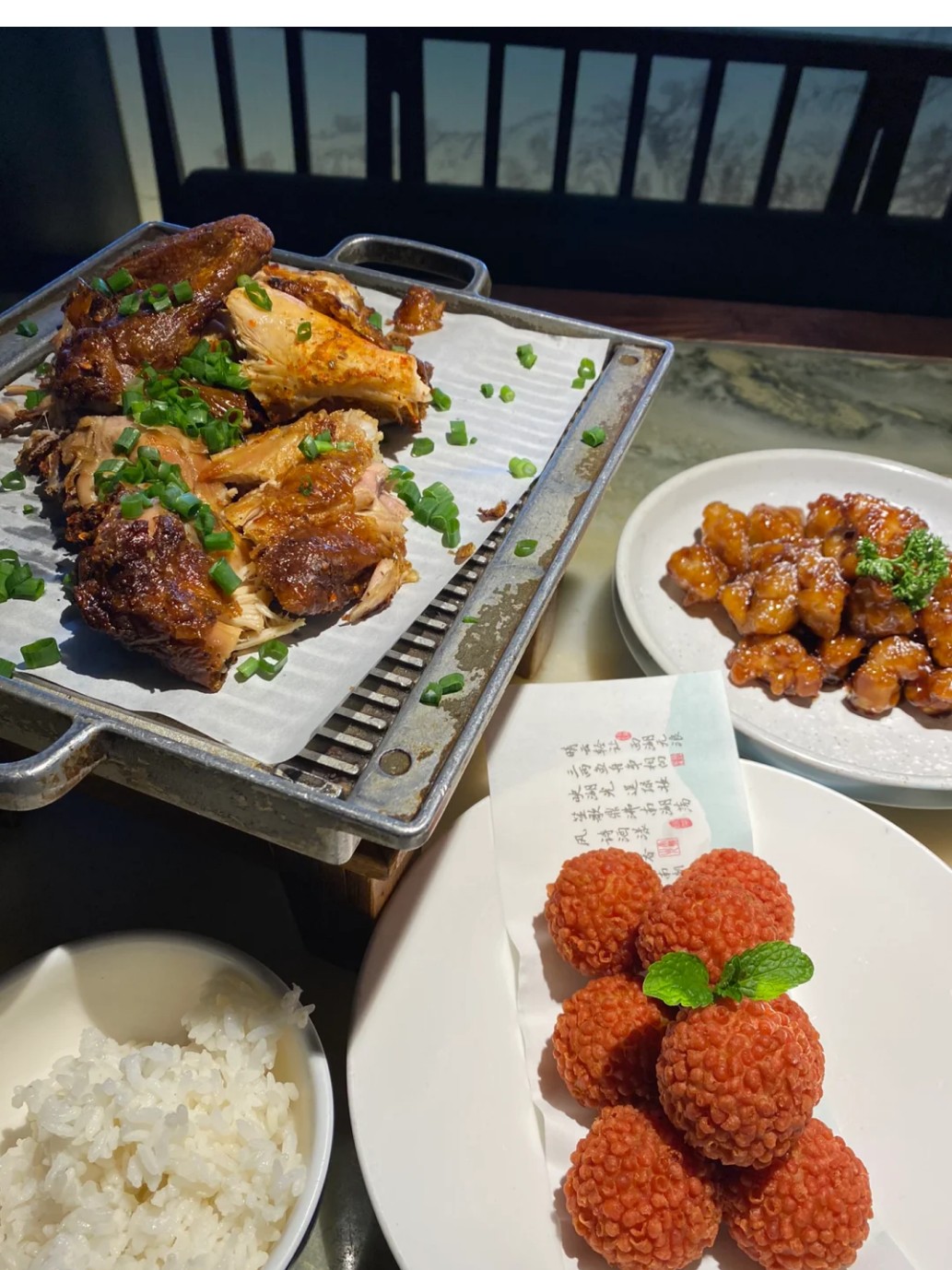

Figure 25: High resolution version sub-figure (c) of Figure 4.

