# OpenReview forum: "CaughtCheating: Is Your MLLM a Good Cheating Detective? Exploring the Boundary of Visual Perception and Reasoning"
_ICLR.cc/2026/Conference — ICLR 2026 Conference Withdrawn Submission_

### Official Review · Reviewer_pwvH · 2025-10-29

**Soundness:** 2
**Presentation:** 2
**Contribution:** 2
**Rating:** 2
**Confidence:** 4

**Summary:**

The paper introduces CaughtCheating, a new benchmark designed to challenge advanced multi-modal large language models (MLLMs) like GPT-o3, which have great performance on existing benchmarks. Inspired by real-world “detective” scenarios from social media, where users analyze photos for subtle clues of suspicious behavior—the benchmark tests models’ ability to integrate fine-grained visual cues with contextual reasoning. Through experiments, the authors find that while GPT-o3 performs well on many hard tasks, it fails almost completely on CaughtCheating scenarios. The study highlights current MLLMs’ limitations in nuanced perceptual reasoning and argues that success on such tasks would mark a step toward human-level detective-like understanding.

**Strengths:**

1. The paper introduces CaughtCheating, a creative and practically relevant benchmark that targets subtle, context-dependent visual reasoning—an underexplored yet important aspect of real-world perception tasks.

2.The study systematically evaluates advanced MLLMs’ visual reasoning processes, revealing specific scenarios where even top-tier models like GPT-o3 fail, thus providing valuable diagnostic insight into current model limitations.

3. By connecting model failures to Guided Search theory, the paper offers an interpretable explanation of why MLLMs struggle, contributing both conceptual depth and guidance for future benchmark and model design.

**Weaknesses:**

1. Section 2.1 provides an informative overview of how modern MLLMs perform visual reasoning. However, the section is unnecessarily verbose (particularly lines 148–160) and contains excessive descriptions of specific model responses that distract from the main narrative. It would be more effective to distill these observations into general patterns or response tendencies across mainstream MLLMs, in line with the section title “Exploring the Boundary of Visual Perception and Reasoning.” Moreover, focusing on a single closed-source model limits the robustness and generality of the analysis.

2. The examples used to demonstrate task difficulty and model behavior are not entirely convincing. For instance, in Figure 4(a), the supposed clue (a finger) appears nearly impossible for a human to detect, raising concerns about the fairness or clarity of the ground truth. Meanwhile, GPT’s response to Figure 4(b) seems reasonable, yet it is judged as incorrect, suggesting potential ambiguity or inconsistency in the labeling criteria.

3. The dataset’s narrow focus on “catching cheating” scenarios undermines its generalizability. While the paper claims to assess general visual reasoning abilities, the data distribution is highly specific and socially contextualized. As such, conclusions such as “failing to identify deterministic clues” are not well supported by a sufficiently diverse or representative sample of reasoning tasks.

**Questions:**

Weakness mainly reflects my concerns.

some typos:
1. Line 037 : unnecessary line breaks
2. Line 091-092: `` detection-level'' should be ``detective-level'' as mentioned before.

---

> ### Author Response · Authors · 2025-11-21
> **Response to Reviewer pwvH**
>
> We sincerely appreciate the time and effort the reviewers took to evaluate our manuscript and provide valuable feedback.
>
> ---
>
> > W1: Section 2.1 provides an informative overview of how modern MLLMs perform visual reasoning. However, the section is unnecessarily verbose (particularly lines 148–160) and contains excessive descriptions of specific model responses that distract from the main narrative. It would be more effective to distill these observations into general patterns or response tendencies across mainstream MLLMs, in line with the section title “Exploring the Boundary of Visual Perception and Reasoning.” Moreover, focusing on a single closed-source model limits the robustness and generality of the analysis.
>
>
> We sincerely thank the reviewer for the suggestions. Our goal is to **provide a detailed analysis of scenarios where the strongest model, o3, succeeds or fails, in order to introduce our task and the corresponding cognitive theory used to explain the sources of difficulty**. We acknowledge the section is currently verbose and will revise it to be more concise.
>
> The primary reason we mainly focus on GPT-o3 for examples and analysis is that, at present, most open-sourced models perform very poorly in this challenging real-world scenario. As shown in Table 1, most open-sourced models achieve an overall F1 of less than 10%. Since these models can hardly generate useful clues for analysis, we primarily focus on o3 as the representative example. We will provide additional case studies in the appendix in future versions.
>
> ---
>
> > W2: The examples used to demonstrate task difficulty and model behavior are not entirely convincing. For instance, in Figure 4(a), the supposed clue (a finger) appears nearly impossible for a human to detect, raising concerns about the fairness or clarity of the ground truth. Meanwhile, GPT’s response to Figure 4(b) seems reasonable, yet it is judged as incorrect, suggesting potential ambiguity or inconsistency in the labeling criteria.
>
>
> We sincerely thank the reviewer for the comments.
> 1. We have added a high-resolution version of the Figure 4 images in the appendix for your reference.
> 2. In (a), as you noted, the fingers are indeed present in the image, and **their existence clearly indicates there is another person in the room, which directly conflicts with the claim of being alone**.
> 3. Regarding GPT’s response to Figure 4(b), **it is actually judged as correct, as indicated by the green check symbol**.
>
> Moreover, to reduce subjectivity in the Real Subset, each image was jointly reviewed by multiple human annotators, who discussed each case, voted on the deterministic clue, and cross-checked their interpretations against comments from the original social media posts. While some subjectivity is unavoidable in real-world scenarios, our protocol minimizes ambiguity by retaining only cases with clear consensus, requiring that clues be visually verifiable, and constraining all questions within a fixed social context to avoid interpretive drift. Finally, thousands of posts and images were manually screened, and only these samples were selected.
>
> ---
>
> > W3: The dataset’s narrow focus on “catching cheating” scenarios undermines its generalizability. While the paper claims to assess general visual reasoning abilities, the data distribution is highly specific and socially contextualized. As such, conclusions such as “failing to identify deterministic clues” are not well supported by a sufficiently diverse or representative sample of reasoning tasks.
>
> We sincerely thank the reviewer for the comments. However, we would like to clarify that **we do NOT claim that our task is for assessing general visual reasoning abilities**, but rather to evaluate performance on a specific, extremely challenging real-world scenario. Our task is distinctive in that **even the strongest o3 model achieves only about 20% performance, highlighting the practical challenges of real-world**, detective-level reasoning. In contrast to many existing benchmarks—where GPT-o3-level models can achieve high performance—we present a real-world scenario that current LLMs are not yet able to solve. Furthermore, our contribution includes a detailed **analysis grounded in cognitive theories, Guided Search, which provides valuable insights as we bridge the gap between LLM benchmarks and well-established cognitive frameworks**.

---

> > ### Comment · Reviewer_pwvH · 2025-11-26
> >
> > Thanks for the clarification. I have several additional questions.
> >
> > 1. Given that these images are quite large in size, could you share the evaluation settings to have table 1? For example, performances of models like Qwen-VL-series vary a lot under different "max_pixels". This also applies to "media_resolution" for Gemini models.
> >
> > 2. I understand the motivation to propose a visual search benchmark for Multimodal LLMs. It is preferred to discuss the difference between existing works like $V^*$[1].
> >
> > [1]https://vstar-seal.github.io/

---

### Official Review · Reviewer_usGt · 2025-10-29

**Soundness:** 2
**Presentation:** 3
**Contribution:** 2
**Rating:** 4
**Confidence:** 4

**Summary:**

The paper introduces CaughtCheating, a new benchmark designed to test whether advanced multi-modal LLMs possess detective-level visual perception and reasoning skills. Motivated by real social-media scenarios where users ask others to find subtle clues contradicting a partner’s claim, the benchmark focuses on identifying extremely inconspicuous, context-dependent visual evidence, such as reflections, small objects, or social inconsistencies, that current models often overlook. The authors analyze why even state-of-the-art models like GPT-o3 fail on these tasks, drawing on Guided Search theory to show that the clues have low bottom-up salience, lack top-down guidance, and require nuanced social interpretation. They construct both a carefully curated real subset and a diverse synthetic subset, and provide automatic evaluation procedures. Experiments reveal stark limitations: even the strongest models achieve only modest accuracy on real-world cases, frequently hallucinating suspicious clues or missing decisive ones.

**Strengths:**

- The dataset captures subtle, real-world visual clues that existing benchmarks overlook, providing a meaningful test of advanced multimodal reasoning.
- The analysis identifies why state-of-the-art models fail, linking errors to low-salience signals and lacking top-down guidance, offering actionable insights for future model design.
- The benchmark includes both real and synthetic subsets, plus automatic scoring procedures, enabling rigorous and scalable assessment of MLLM capabilities.

**Weaknesses:**

- The benchmark focuses on a specific “detective-style clue finding” scenario, which may be partially included in other work and limits its generalizability to other multimodal reasoning tasks.
- Some clues in real-world images may be open to interpretation, making the correctness labels somewhat subjective.
- The prompt design is too simple, which cannot cover enough requirements in the real world for fine-grained perception and reasoning.

**Questions:**

- How were ambiguous or borderline “suspicious clues” handled during annotation, and what measures ensured consistency across annotators?
- Could the task design unintentionally encourage models to hallucinate suspicious clues due to its “detective-style” framing?
- Did you study whether models improve with more explicit top-down instructions (e.g., “look for reflections”)?

---

> ### Author Response · Authors · 2025-11-21
> **Response to Reviewer usGt**
>
> We sincerely appreciate the time and effort the reviewers took to evaluate our manuscript and provide valuable feedback.
>
> ---
>
> > W1: The benchmark focuses on a specific “detective-style clue finding” scenario, which may be partially included in other work and limits its generalizability to other multimodal reasoning tasks.
>
>
> We sincerely thank the reviewer for the comments. We have updated our manuscript by **adding an additional section to the Related Work**, where we compare several recent concurrent works that discuss detective-level tasks. However, to the best of our knowledge, there is currently no benchmark that explicitly addresses scenarios like ours. The most relevant prior works are [1] and [2]: [1] represents a geoguessr-style task, which is a fundamentally different setting, while [2] includes some more general scenarios. Our task is distinctive in that **even the strongest o3 model achieves only about 20% performance, highlighting the practical challenges of real-world**, detective-level reasoning. In contrast to many existing benchmarks—where GPT-o3-level models can achieve high performance—we present a real-world scenario that current LLMs are not yet able to solve. Furthermore, our contribution includes a detailed **analysis grounded in cognitive theories, Guided Search, which provides valuable insights as we bridge the gap between LLM benchmarks and well-established cognitive frameworks**.
>
> [1] Luo, W., Lu, T., Zhang, Q., Liu, X., Hu, B., Zhao, Y., ... & Xiao, C. (2025). Doxing via the Lens: Revealing Location-related Privacy Leakage on Multi-modal Large Reasoning Models. arXiv preprint arXiv:2504.19373.
> [2] Liu, F., Zhang, Y., Huang, X., Peng, Y., Li, X., Wang, L., ... & Dong, W. (2025, October). The Eye of Sherlock Holmes: Uncovering User Private Attribute Profiling via Vision-Language Model Agentic Framework. In Proceedings of the 33rd ACM International Conference on Multimedia (pp. 4875-4883).
>
> ---
>
> > W2: Some clues in real-world images may be open to interpretation, making the correctness labels somewhat subjective.
>
>
> We thank the reviewer for the comments. To reduce subjectivity in the Real Subset, each image was jointly reviewed by multiple human annotators. The annotators discussed each case, voted on the most deterministic clue, and cross-checked their interpretations with comments from the original social media posts. While some subjectivity is inevitable in real-world scenarios, our protocol minimizes ambiguity by retaining only cases with clear consensus, requiring that clues be visually verifiable, and constraining all questions to a fixed social context to avoid interpretive drift. Finally, thousands of posts and images were manually screened, and the selected samples reflect this careful curation.
>
> ---
>
> > W3: The prompt design is too simple, which cannot cover enough requirements in the real world for fine-grained perception and reasoning.
>
>
> We thank the reviewer for the question. To generate synthetic data, we provided specific instructions that indicated pre-defined, templated clues. During the synthetic data procedure, we did our best to make the scenarios as balanced as possible.
>
> We agree that clues in the Synthetic Subset may appear more visually prominent than those in real-world scenarios. This is partly due to current limitations in image generation models, which recent studies [3, 4] have shown still struggle to reliably follow complex prompts—especially those involving multiple objects, fine-grained details, and spatial relationships. As a result, **these models often fail to produce the subtle, ambiguous cues typically present in real-world settings (e.g., reflections)**. As the first work focusing on this scenario, generation with generative models is intended as an exploration of scaling up data. However, the difficulty in generating these types of images **further underscores the value of our collected real data and tasks, which remain practical real-world scenarios that current VLMs cannot yet solve**.
>
> [3] Sella, E., Kleiman, Y., & Averbuch-Elor, H. (2025, August). InstanceGen: Image Generation with Instance-level Instructions. In Proceedings of the Special Interest Group on Computer Graphics and Interactive Techniques Conference Conference Papers (pp. 1-10).
>
> [4] Zhou, Y., Yuan, J., & Wang, Q. (2025). Draw ALL Your Imagine: A Holistic Benchmark and Agent Framework for Complex Instruction-based Image Generation. arXiv preprint arXiv:2505.24787.

---

> > ### Author Response · Authors · 2025-11-21
> >
> > > Q1: How were ambiguous or borderline “suspicious clues” handled during annotation, and what measures ensured consistency across annotators?
> >
> > Please kindly refer to W2.
> >
> > ---
> >
> > > Q2: Could the task design unintentionally encourage models to hallucinate suspicious clues due to its “detective-style” framing?
> >
> >
> > We thank the reviewer for the question. In our task, we have explicitly designed the evaluation suites to mitigate hallucination:
> >
> > 1. Use of Unclued images: By introducing Unclued images, we penalize models for hallucination. If a model generates clues on Unclued images, its accuracy will be low. By separately reporting performance on Clued and Unclued images, we can easily analyze how different models behave. For example, while o3 is good at finding clues, it is also prone to hallucinate.
> >
> > 2. Use of IoU on Clued images: Even for Clued images, we discourage hallucination by using the IoU metric. If VLMs generate too many unrelated clues, their IoU scores will be low.
> >
> > In summary, our evaluation metrics are specifically designed to reduce and detect hallucinations.
> >
> >
> > ---
> >
> > > Q3: Did you study whether models improve with more explicit top-down instructions (e.g., “look for reflections”)?
> >
> > We thank the reviewer for the valuable comments. Our analysis concludes that the primary bottleneck in current VLMs lies in their reasoning capability. While they can perceive abnormal parts if we point them out specifically, they cannot infer their significance. For example, as shown in Figure 1, even when VLMs are able to identify the reflection in the spoons, they are **unable to construct the latent logical chain that "the reflection suggests the presence of someone else, which conflicts with the claim of being alone."**
> >
> >
> > Based on the above analysis, we conducted a further experiment in which we **collected all the potential clues in our data, explicitly stated the meaning of each clue, and provided this information as part of the prompts for GPT-o3**. As shown below, we observed a substantial improvement in o3's performance. This experiment serves as a proof-of-concept that assisting VLMs in building the underlying logical connections can indeed improve their performance.
> >
> > | Models| Clued Acc | Clued IoU | Unclued Acc | Overall F1|
> > | ---- | ---- |---- | ---- | ---- |
> > | GPT-o3 (ori)| 26.0 | 17.2 | 8.0 | 23.9 |
> > | GPT-o3 (clues in prompt)| 52.0 | 20.7 | 36.0 | 40.7 |
> >
> >
> > Moreover, we also tried the finetuning method described in Q3 of reviewer vboB. Although this improves the performance of open-sourced VLMs, these models become prone to generating clues prematurely.

---

### Official Review · Reviewer_VCFg · 2025-10-29

**Soundness:** 3
**Presentation:** 2
**Contribution:** 2
**Rating:** 4
**Confidence:** 4

**Summary:**

The paper introduces CaughtCheating, a benchmark for detective visual perception with annotated clues and decomposed questions. It aims to find subtle, context-dependent clues in images that contradict the given claim. Using this benchmark, state-of-the-art models perform poorly, suggesting a gap between current MLLMs and detection capabilities.

**Strengths:**

1.	Interesting, socially relevant task that focuses on subtle visual perception and context-sensitive reasoning on image-claim pairs.
2.	The annotation schema distinguishes deterministic and non-deterministic clues in the detective task and decomposes visual reasoning questions, aiming to evaluate the performance of MLLMs.

**Weaknesses:**

1.	The scale of the constructed real dataset is quite small, with only 100 samples. A small real dataset raises concerns about the reliability of the benchmarking results and also weakens the contributions of benchmarking.
2.	The paper primarily constructs a benchmark and reports the limitations of current MLLMs. It does not discuss the potential direction or insightful methods to address the identified failures.
3.	Synthetic subset generation relies on stereotypes and templated clues. The paper acknowledges that it is easier, which can distort model comparisons and may encode social biases.
4.	Results on real and synthetic datasets are significantly different, raising concerns about whether the proposed dataset can benchmark current methods and whether it can truly improve the performance of current methods.

**Questions:**

1.	See above.
2.	Can you report breakdowns by scene, gender presentation, and cue types to identify biased failure modes?

---

> ### Author Response · Authors · 2025-11-21
> **Response to Reviewer VCFg**
>
> We sincerely appreciate the time and effort the reviewers took to evaluate our manuscript and provide valuable feedback.
>
> ---
>
> >  W1: The scale of the constructed real dataset is quite small, with only 100 samples. A small real dataset raises concerns about the reliability of the benchmarking results and also weakens the contributions of benchmarking.
>
> We acknowledge that the Real subset appears relatively small in size; however, this is attributable to both the **inherent scarcity and difficulty** of such examples, and our **rigorous standards** for their collection:
> 1. Scarcity & Difficulty: This scenario arises when clues are **so subtle that even the original owners of the photos overlook them**. Therefore, this task has practical value, as it directly assesses whether VLMs can detect clues that humans cannot, representing a practical real-world scenario that requires detective-level capabilities.
> 2. High Standards: For each image, we conducted thorough investigations based on feedback from every annotator and from the original public comments to determine the existence of clues. Thousands of posts and images were manually screened before selecting these samples.
>
> Moreover, we would like to emphasize that this task represents a practical real-world scenario that requires detective-level capabilities. Unlike most existing benchmarks, where GPT-o3-level models achieve rather high performance, **our proposed scenario remains unsolved by current LLMs**. Furthermore, our contribution includes a detailed **analysis grounded in cognitive theories, Guided Search, which provides valuable insights as we bridge the gap between LLM benchmarks and well-established cognitive frameworks**.
>
> ---
>
> > W2: The paper primarily constructs a benchmark and reports the limitations of current MLLMs. It does not discuss the potential direction or insightful methods to address the identified failures.
>
>
> We thank the reviewer for the valuable comments. In our original paper, we not only (i) analyze the reasoning traces of VLMs, but also (ii) introduce the **Guided Search theory to investigate why our task is inherently difficult**, and (iii) **decompose the visual perception and reasoning components to pinpoint where VLMs make errors**.
>
> Our analysis concludes that the primary bottleneck in current VLMs lies in their reasoning capability. While they can perceive abnormal parts if we point them out specifically, they cannot infer their significance. For example, as shown in Figure 1, even when VLMs are able to identify the reflection in the spoons, they are **unable to construct the latent logical chain that "the reflection suggests the presence of someone else, which conflicts with the claim of being alone."**
>
>
> Based on the above analysis, we conducted a further experiment in which we **collected all the potential clues in our data, explicitly stated the meaning of each clue, and provided this information as part of the prompts for GPT-o3**. As shown below, we observed a substantial improvement in o3's performance. This experiment serves as a proof-of-concept that assisting VLMs in building the underlying logical connections can indeed improve their performance.
>
> | Models| Clued Acc | Clued IoU | Unclued Acc | Overall F1|
> | ---- | ---- |---- | ---- | ---- |
> | GPT-o3 (ori)| 26.0 | 17.2 | 8.0 | 23.9 |
> | GPT-o3 (clues in prompt)| 52.0 | 20.7 | 36.0 | 40.7 |
>
>
> Moreover, we also tried the finetuning method described in Q3 of reviewer vboB. Although this improves the performance of open-sourced VLMs, these models become prone to generating clues prematurely.
>
> ---
>
> > W3: Synthetic subset generation relies on stereotypes and templated clues. The paper acknowledges that it is easier, which can distort model comparisons and may encode social biases.
>
> We thank the reviewer for the question. To generate synthetic data, we provided specific instructions that included the use of pre-defined templated clues. Throughout the synthetic data generation process, we made every effort to ensure the scenarios were as balanced as possible.
>
> ---

---

> > ### Author Response · Authors · 2025-11-21
> >
> > > W3: Synthetic subset generation relies on stereotypes and templated clues. The paper acknowledges that it is easier, which can distort model comparisons and may encode social biases.
> >
> > We thank the reviewer for the question. To generate synthetic data, we provided specific instructions that included the use of pre-defined templated clues. Throughout the synthetic data generation process, we made every effort to ensure the scenarios were as balanced as possible.
> >
> > ---
> >
> > > W4: Results on real and synthetic datasets are significantly different, raising concerns about whether the proposed dataset can benchmark current methods and whether it can truly improve the performance of current methods.
> >
> > We sincerely thank the reviewer for the comments. We agree that clues in the Synthetic Subset may appear more visually prominent than those in real-world scenarios. This is partly due to current limitations in image generation models, which recent studies [1, 2] have shown still struggle to reliably follow complex prompts—especially those involving multiple objects, fine-grained details, and spatial relationships. As a result, **these models often fail to produce the subtle, ambiguous cues typically present in real-world settings (e.g., reflections)**. As the first work focusing on this scenario, generation with generative models is intended as an exploration of scaling up data. However, the difficulty in generating these types of images **further underscores the value of our collected real data and tasks, which remain practical real-world scenarios that current VLMs cannot yet solve**.
> >
> > [1] Sella, E., Kleiman, Y., & Averbuch-Elor, H. (2025, August). InstanceGen: Image Generation with Instance-level Instructions. In Proceedings of the Special Interest Group on Computer Graphics and Interactive Techniques Conference Conference Papers (pp. 1-10).
> >
> > [2] Zhou, Y., Yuan, J., & Wang, Q. (2025). Draw ALL Your Imagine: A Holistic Benchmark and Agent Framework for Complex Instruction-based Image Generation. arXiv preprint arXiv:2505.24787.
> >
> > ---
> >
> > > Q2: Can you report breakdowns by scene, gender presentation, and cue types to identify biased failure modes?
> >
> > We sincerely thank the reviewer for the valuable suggestion. We present the model results across different subsets as shown below:
> >
> > | model_name        | Dining (clue_acc) | Dining (unclue_acc) | Hotel (clue_acc) | Hotel (unclue_acc) | Male (clue_acc) | Male (unclue_acc) | Female (clue_acc) | Female (unclue_acc) |
> > |-------------------|-------------------|----------------------|------------------|---------------------|------------------|--------------------|--------------------|-----------------------|
> > | InternVL2.5-40b   | 0.0               | 44.4                 | 2.6              | 90.3                | 0.0              | 73.7               | 5.9                | 73.3                 |
> > | Gemini2-flash     | 18.2              | 5.6                  | 7.9              | 6.5                 | 5.9              | 10.5               | 17.7               | 3.3                  |
> > | Gemini2.5-flash   | 27.3              | 11.1                 | 13.2             | 29.0                | 14.7             | 31.6               | 23.5               | 16.7                 |
> > | Gemini2.5-Pro     | 18.2              | 11.1                 | 18.4             | 32.3                | 8.8              | 10.5               | 35.3               | 33.3                 |
> > | GPT-o3            | 36.4              | 0.0                  | 21.1             | 12.9                | 20.6             | 5.3                | 35.3               | 10.0                 |
> >
> >
> > As shown in the table, models have relatively higher performance on the Dining scene compared to Hotel, and also perform better on Female than Male. However, there is no significant bias between categories. Our current conclusion remains that existing models have limited capability in handling this task.

---

### Official Review · Reviewer_vboB · 2025-10-31

**Soundness:** 2
**Presentation:** 3
**Contribution:** 3
**Rating:** 6
**Confidence:** 4

**Summary:**

This paper addresses the gap in evaluating detective-level visual perception and social reasoning in MLLMs by proposing **CaughtCheating**, a benchmark inspired by real-world social media requests to detect suspicious clues contradicting a claim. The benchmark includes a Real Subset (100 manually curated social media images, split into Clued/Unclued) and a Synthetic Subset (3700 GPT-Image-1 generated images) to balance realism and scalability. Grounded in cognitive science’s Guided Search theory, the paper explains MLLM failures: clues have low bottom-up salience, lack top-down feature guidance, and require social context interpretation—rendering MLLMs’ exhaustive search ineffective. Extensive experiments on 17 MLLMs (open/closed-source) show top models like GPT-o3 achieve only 26.0% Clued Accuracy, 17.2% IoU, 8.0% Unclued Accuracy, and 23.9% F1 on the Real Subset, exposing limitations like hallucination in unclued images. Contributions include the benchmark, theoretical failure analysis, and insights into MLLMs’ detective capability boundaries.

**Strengths:**

* The paper proposes a genuinely novel and highly challenging task that pushes MLLMs beyond standard VQA or reasoning benchmarks.

- The benchmark is thoughtfully constructed with two complementary subsets: a small, extremely difficult Real Subset that captures real-world subtlety and a large-scale Synthetic Subset for diversity. The annotation scheme is very thorough, distinguishing between "Deterministic" and "Non-deterministic" clues, and including "Decomposed" (Perception vs. Reasoning) questions, which enables a fine-grained failure analysis.

- The paper provides crucial insights into SOTA model failures. The key finding is twofold: models are not only bad at finding the *correct* clue (e.g., 26.0% Clued Acc for o3) but are *also* prone to hallucinating *incorrect* clues on innocent images (e.g., 8.0% Unclued Acc for o3). The decomposition analysis (Table 2) brilliantly supports the theoretical claims by showing that models *can* perceive the clues if prompted (high Dec. P Acc) but fail in the *search and discovery* process.

**Weaknesses:**

* The Real Subset (100 images) is small, dominated by hotel scenes (69%), and biased toward cisgender heterosexual couples—failing to represent diverse scenarios (e.g., workplaces) or relationships, which limits conclusions about MLLMs’ performance across real-world contexts.
* Clues in the Synthetic Subset are overly prominent (e.g., obvious lipstick, two glasses) due to prompt design, probably making it less effective at mimicking the subtlety of real-world clues (e.g., faint reflections) and reducing its value for training/testing MLLMs on authentic challenges.
* The authors rightly acknowledge the ethical limitations in Section 7. The "Real Subset" is sourced from public data that overwhelmingly represents cisgender, heterosexual couples. While the synthetic data attempts to balance gender, the entire benchmark is framed around a single, socially charged, and potentially harmful application (detecting infidelity). This narrow, biased framing limits the generalizability of the "detective-level" reasoning claims.
* The data and code not be publicly released.

**Questions:**

* Can you expand the Real Subset to include more scenes (e.g., offices, public transit) to address current scenes' biases?
* Did you measure difficulty (e.g., inter-model accuracy variance, human annotation time) across dimensions/sub-fields? Are there imbalances (e.g., over-representation of easy Daily Life questions) that might skew model rankings?
* Have you fine-tuned MLLMs on the CaughtCheating dataset (especially the Real Subset) to test if it improves their detective capabilities?
* Advanced models like GPT-o3 hallucinate more on unclued images—have you analyzed their reasoning traces to identify root causes?

I look forward to an active discussion with the authors during the rebuttal phase and will revise my score accordingly.

**Details Of Ethics Concerns:**

The real dataset used in this paper comes from controversial images uploaded by others on the internet. Although the authors adopted methods to remove privacy in the paper, I personally have ethical concerns because the dataset has not yet been open-sourced.

---

> ### Author Response · Authors · 2025-11-21
> **Response to Reviewer vboB**
>
> We sincerely appreciate the time and effort the reviewers took to evaluate our manuscript and provide valuable feedback.
>
> ---
>
> > W1.1: The Real Subset (100 images) is small
>
> We acknowledge that the Real subset appears relatively small in size; however, this is attributable to both the **inherent scarcity and difficulty** of such examples, and our **rigorous standards** for their collection:
> 1. Scarcity & Difficulty: This scenario arises when clues are **so subtle that even the original owners of the photos overlook them**. Therefore, this task has practical value, as it directly assesses whether VLMs can detect clues that humans cannot, representing a practical real-world scenario that requires detective-level capabilities.
> 2. High Standards: For each image, we conducted thorough investigations based on feedback from every annotator and from the original public comments to determine the existence of clues. Thousands of posts and images were manually screened before selecting these samples.
>
> ---
>
> > W1.2: dominated by hotel scenes (69%), and biased toward cisgender heterosexual couples
>
> Regarding scenario diversity and heterosexuality, our selection is guided by practical concerns:
> 1. Scenario diversity: It is difficult to determine if two people sharing a room in a workplace environment have an unusual relationship. To avoid making potentially unwarranted conclusions, we prioritized more clear-cut scenarios, such as hotels.
> 2. Heterosexuality: Similarly, it is challenging to infer abnormal situations involving two people of the same gender, even if they are observed sharing a hotel room or bed. Therefore, we chose to avoid such assumptions in our data selection.
>
> ---
>
> > W2: Clues in the Synthetic Subset are overly prominent (e.g., obvious lipstick, two glasses) due to prompt design, probably making it less effective at mimicking the subtlety of real-world clues (e.g., faint reflections) and reducing its value for training/testing MLLMs on authentic challenges.
>
> We sincerely thank the reviewer for the comments. We agree that clues in the Synthetic Subset may appear more visually prominent than those in real-world scenarios. This is partly due to current limitations in image generation models, which recent studies [1, 2] have shown still struggle to reliably follow complex prompts—especially those involving multiple objects, fine-grained details, and spatial relationships. As a result, **these models often fail to produce the subtle, ambiguous cues typically present in real-world settings (e.g., reflections)**. As the first work focusing on this scenario, generation with generative models is intended as an exploration of scaling up data. However, the difficulty in generating these types of images **further underscores the value of our collected real data and tasks, which remain practical real-world scenarios that current VLMs cannot yet solve**.
>
> [1] InstanceGen: Image Generation with Instance-level Instructions. In Proceedings of the Special Interest Group on Computer Graphics and Interactive Techniques Conference Conference Papers (pp. 1-10).
>
> [2] Draw ALL Your Imagine: A Holistic Benchmark and Agent Framework for Complex Instruction-based Image Generation. arXiv preprint arXiv:2505.24787.
>
> ---
>
> > W3: The authors rightly acknowledge the ethical limitations in Section 7. The "Real Subset" is sourced from public data that overwhelmingly represents cisgender, heterosexual couples. While the synthetic data attempts to balance gender, the entire benchmark is framed around a single, socially charged, and potentially harmful application (detecting infidelity). This narrow, biased framing limits the generalizability of the "detective-level" reasoning claims.
>
>
> We sincerely thank the reviewer for the comments. Please kindly refer to W1.2 for the diversity concern.
> As mentioned in W1.1, we follow **strict standards when selecting real images**: each example must contain clear, visually verifiable evidence of the target activity without relying on external context. This criterion necessarily restricts the types of images that can be included. Additionally, we aim to **ensure that the selected cases are genuinely challenging, even for human evaluators**. For example, detecting two individuals leaning together as seen through the reflection on a metallic spoon exemplifies the level of difficult yet feasible evidence we aim to include.
>
> Regarding the claim of "detective-level" reasoning capability, we would like to clarify that this task indeed represents a practical, real-world scenario requiring such advanced reasoning skills. Unlike most existing benchmarks, where GPT-o3-level models achieve rather high performance, **our proposed scenario remains unsolved by current LLMs**. Furthermore, our contribution includes a detailed **analysis grounded in cognitive theories, Guided Search, which provides valuable insights as we bridge the gap between LLM benchmarks and well-established cognitive frameworks**.

---

> > ### Author Response · Authors · 2025-11-21
> >
> > > W4: The data and code not be publicly released.
> >
> > We sincerely than the reviewer for the interets in our data. We would like to clarify that both the data and code have been released. A repository link (following double-blind requirements) is provided in the abstract of the pdf.
> >
> > ---
> >
> > > Q1 Can you expand the Real Subset to include more scenes (e.g., offices, public transit) to address current scenes' biases?
> >
> > We thank the reviewer for this valuable suggestion. We agree that expanding the Real Subset to include a wider variety of scenes would be beneficial. We will continue to pursue such extensions; however, collecting real-world images that provide visually verifiable evidence and meet our strict standards remains extremely challenging. Nevertheless, we will strive to broaden our coverage as much as possible and will further discuss this limitation in the paper.
> >
> > ---
> >
> > > Q2: Did you measure difficulty (e.g., inter-model accuracy variance, human annotation time) across dimensions/sub-fields? Are there imbalances (e.g., over-representation of easy Daily Life questions) that might skew model rankings?
> >
> > We sincerely thank the reviewer for the valuable suggestion. However, it is difficult to record human annotation time, as many of these samples are also challenging for human evaluators. For samples that are relatively easy, human evaluators need less than 10 seconds to identify the clue. However, for the difficult ones, human evaluators are unable to find the clue until the ground truth is provided.
> >
> > Instead, we present the model results across different subsets as shown below:
> >
> > | model_name        | Dining (clue_acc) | Dining (unclue_acc) | Hotel (clue_acc) | Hotel (unclue_acc) | Male (clue_acc) | Male (unclue_acc) | Female (clue_acc) | Female (unclue_acc) |
> > |--------|--------|---------|-------|--------|------------|--------------------|--------------------|-----------------------|
> > | InternVL2.5-40b   | 0.0               | 44.4                 | 2.6              | 90.3                | 0.0              | 73.7               | 5.9                | 73.3                 |
> > | Gemini2-flash     | 18.2              | 5.6                  | 7.9              | 6.5                 | 5.9              | 10.5               | 17.7               | 3.3                  |
> > | Gemini2.5-flash   | 27.3              | 11.1                 | 13.2             | 29.0                | 14.7             | 31.6               | 23.5               | 16.7                 |
> > | Gemini2.5-Pro     | 18.2              | 11.1                 | 18.4             | 32.3                | 8.8              | 10.5               | 35.3               | 33.3                 |
> > | GPT-o3            | 36.4              | 0.0                  | 21.1             | 12.9                | 20.6             | 5.3                | 35.3               | 10.0                 |
> >
> >
> > As shown in the table, models have relatively higher performance on the Dining scene compared to Hotel, and also perform better on Female than Male. However, there is no significant bias between categories. Our current conclusion remains that existing models have limited capability in handling this task.
> >
> > ---
> >
> > > Q3: Have you fine-tuned MLLMs on the CaughtCheating dataset (especially the Real Subset) to test if it improves their detective capabilities?
> >
> > We thank the reviewer for this valuable suggestion. We finetuned Qwen2.5-VL-7B using a 10-fold cross-validation method. Specifically, we split the Real set into 10 subsets and finetuned Qwen2.5-VL-7B 10 times, each time using 90 samples for training (with human-labeled groundtruth) and 10 for testing. The following table reports the average performance across all folds:
> >
> > | Models | Clued Acc | Clued IoU | Unclued Acc | Overall F1 |
> > |-----------|-----------|-----------|-------------|------------|
> > | Qwen2.5-VL-7B (ori) |   2.0    |    3.9   |     66.0     |   2.9     |
> > | Qwen2.5-VL-7B (finetuned) |   18.0    |    3.7    |     20.0     |   15.2     |
> >
> > As shown above, **finetuning on this dataset enables Clued Acc to increase significantly from nearly zero to 18.0**, reaching the level of Gemini2.5-flash. However, both Clued IoU and Unclued Acc do not improve substantially. After a thorough analysis of the model responses, we found that finetuning on this data leads to **more false negatives**. In particular, after finetuning, the **LLMs tend to generate clues prematurely**. This is an undesirable outcome, which further highlights the challenges inherent in our task.
> >
> > ---
> >
> > > Q4: Advanced models like GPT-o3 hallucinate more on unclued images—have you analyzed their reasoning traces to identify root causes?
> >
> >
> > Yes, we also carefully analyzed the reasoning traces when GPT-o3 was presented with unclued images. As illustrated in Figure 22, we observed that when prompted to identify something suspicious, the model tends to systematically scan every part of the image and mark almost everything as suspicious, without truly interpreting the underlying meaning.

---

> > > ### Comment · Reviewer_vboB · 2025-11-27
> > > **Response to the author**
> > >
> > > Thanks for the detailed response. I have read the author's rebuttal carefully, along with the other reviews, and most of my concerns have been resolved. However, I still think the author should expand the data and diverse scenarios in the Real Subset (100 images seem too small). Given the paper's scope and experiments, I remain mostly positive and will keep my scores.

---

### Note · Authors · 2026-01-05

I have read and agree with the venue's withdrawal policy on behalf of myself and my co-authors.